# Generalized Category Discovery under Domain Shift: A Frequency Domain Perspective

**Wei Feng**[1,2*]   **Zongyuan Ge**[1,2]
[1]Monash University, Clayton, VIC 3800, Australia
[2]Airdoc–Monash Research, Monash University, Clayton, VIC 3800, Australia
`wf02429@gmail.com`   `zongyuan.ge@monash.edu`

## Abstract

Generalized Category Discovery (GCD) aims to leverage labeled samples from known categories to cluster unlabeled data that may include both known and unknown categories. While existing methods have achieved impressive results under standard conditions, their performance often deteriorates in the presence of distribution shifts. In this paper, we explore a more realistic task: Domain-Shifted Generalized Category Discovery (DS_GCD), where the unlabeled data includes not only unknown categories but also samples from unknown domains. To tackle this challenge, we propose a **F**requency-guided Gene**r**alized Cat**e**gory Discov**e**ry framework (FREE) that enhances the model's ability to discover categories under distributional shift by leveraging frequency-domain information. Specifically, we first propose a frequency-based domain separation strategy that partitions samples into known and unknown domains by measuring their amplitude differences. We then propose two types of frequency-domain perturbation strategies: a cross-domain strategy, which adapts to new distributions by exchanging amplitude components across domains, and an intra-domain strategy, which enhances robustness to intra-domain variations within the unknown domain. Furthermore, we extend the self-supervised contrastive objective and semantic clustering loss to better guide the training process. Finally, we introduce a clustering-difficulty-aware resampling technique to adaptively focus on harder-to-cluster categories, further enhancing model performance. Extensive experiments demonstrate that our method effectively mitigates the impact of distributional shifts across various benchmark datasets and achieves superior performance in discovering both known and unknown categories.

## 1 Introduction

Deep learning has achieved remarkable success in visual recognition tasks under closed-world assumptions [25, 26, 13]. However, visual concepts in the real world are infinite, open-ended, and constantly evolving, while traditional deep learning models are limited to recognizing a predefined set of categories and cannot handle unseen concepts. In contrast, humans can recognize new concepts based on existing knowledge. Inspired by this, Generalized Category Discovery (GCD) [33, 38] has been proposed to enable models to simultaneously recognize both known and new categories. However, existing GCD methods typically assume that labeled and unlabeled data come from the same domain, which is often not the case in real-world scenarios. In practice, data often exhibit both label and domain shifts. For instance, in clinical applications, medical images collected from different sources may differ significantly in both appearance and statistical properties, and new disease categories may emerge unexpectedly. Current GCD methods suffer substantial performance

---

*Corresponding author.

39th Conference on Neural Information Processing Systems (NeurIPS 2025).

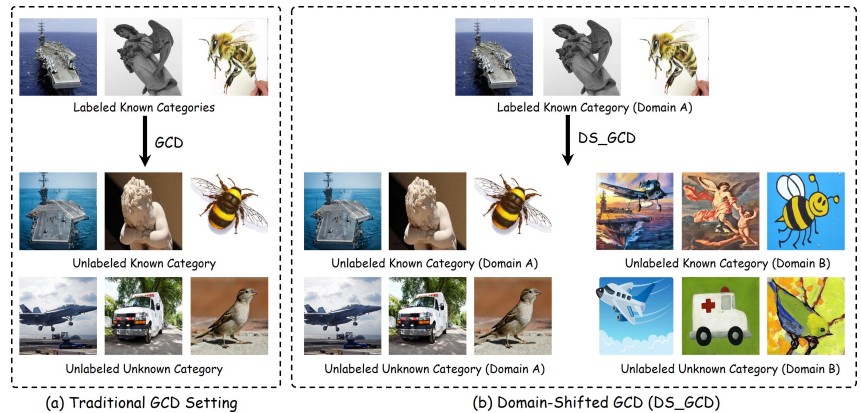

Figure 1: Illustration of Domain-Shifted GCD setting (DS_GCD) and the traditional GCD setting. In the DS_GCD setting, the model needs to categorize known and unknown categories from both known and unknown domains.

degradation when faced with both unknown categories and domain shifts [35]. This work addresses a more realistic and challenging problem setting, referred to as Domain-Shifted Generalized Category Discovery (DS_GCD), where unlabeled data may belong to both unseen categories and entirely new domains, as illustrated in Fig. 1. DS_GCD requires models to identify novel categories and domains using only labeled data from known categories, while generalizing across both semantic and distributional variations. This setting poses major challenges for existing methods. First, most unsupervised domain adaptation (UDA) approaches cannot be directly applied to DS_GCD, as new categories lack label supervision, and relying on unlabeled data often leads to poor performance. Second, current GCD methods lack mechanisms to address domain shifts, resulting in significant performance drops with novel domains.

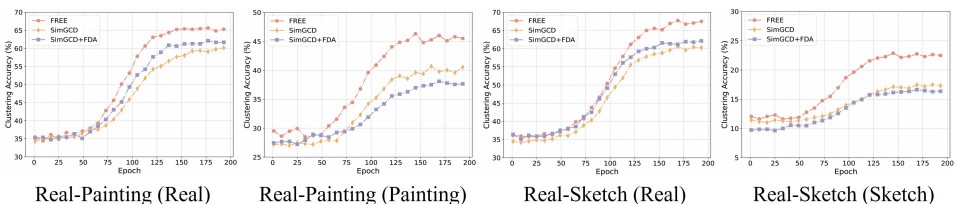

| Real-Painting (Real) | Real-Painting (Painting) | Real-Sketch (Real) | Real-Sketch (Sketch) |

Figure 2: Clustering accuracy of all categories using FREE, SimGCD+FDA, and SimGCD. It can be observed that FREE outperforms SimGCD+FDA both in final clustering accuracy and convergence speed. In contrast, random transformation in SimGCD+FDA may even hinder the learning of unknown domains, potentially leading to negative transfer effects.

To address the aforementioned challenges, we propose **F**requency-guided Gene**r**alized Cat**e**gory Discov**e**ry framework (FREE), a method that tackles the DS_GCD task from a frequency-domain perspective. Our work is inspired by Fourier Domain Adaptation (FDA) [42], which bridges domain gaps by leveraging Fourier Transform to exchange low-frequency components (typically associated with image style) across different domains in the frequency domain. However, directly applying the FDA to the GCD task poses two key challenges. First, FDA assumes that randomly exchanging amplitude components between images from different domains can enhance performance on the target domain. However, in the DS_GCD setting, where domain labels for individual samples are unavailable and the direction of style transfer cannot be controlled, randomly swapping amplitude components between images may lead to unstable training and negative transfer, as shown in Figure 2, especially in the unknown domain. This is because transforming unknown domain samples into the style of the known domain may distort their original structure or semantic content. Unlike the known domain, the unknown domain lacks any supervision and relies solely on pseudo-labels to learn clustering, making the learning process more vulnerable to such instability. Furthermore, this strategy primarily focuses on global domain adaptation while overlooking intra-class style variations across domains. To address these issues, we propose a divide-and-conquer strategy that overcomes these limitations in

a more controlled and class-aware manner. Specifically, we first design a frequency-based domain separation strategy, which separates known-domain and unknown-domain samples based on the amplitude differences across different sample sets. We then propose a cross-domain frequency perturbation strategy that integrates amplitude components from different domains in a class-aware manner to generate hybrid samples and bridge distribution gaps. Additionally, for unknown domains, we introduce an intra-domain frequency perturbation strategy to enhance the model's robustness to domain-specific variations. Furthermore, we extend the self-supervised contrastive objective and semantic clustering loss to better guide the training process. These enhancements enable the model to effectively leverage cross-domain frequency knowledge and discover novel categories from unlabeled images, even under domain shift conditions. Finally, we propose a clustering-difficulty-aware resampling strategy, which dynamically evaluates the clustering difficulty of different categories and encourages the model to focus more on the harder ones. In summary, our contributions are as follows: 1) We introduce a novel frequency-domain approach to address the GCD task under domain shifts. 2) We propose frequency-based domain separation and perturbation strategies to improve domain robustness and cross-domain generalization. 3) We enhance learning with extended contrastive learning and clustering objectives. 4) We develop a clustering-difficulty-aware resampling mechanism to focus learning on challenging categories. 5) Extensive experimental results show that our method is able to remarkably discover new categories while minimizing the effects of domain shift, and performs far better than state-of-the-art methods on all datasets.

## 2   Related Works

**Category Discovery**   Category discovery tasks are typically categorized into Novel Class Discovery (NCD) [5] and Generalized Category Discovery (GCD) [8] settings. NCD was initially introduced by [12], with the goal of clustering unlabeled samples from novel categories using only labeled data from known categories. Early methods primarily relied on ranking statistics and self-supervised contrastive learning to transfer discriminative knowledge from known to unknown categories [12, 44]. Subsequently, UNO [9] utilized a self-labeling strategy based on the Sinkhorn-Knopp (SK) algorithm, integrating multiple objectives into a unified framework to significantly enhance discovery performance. Later works further explored category-level semantic relationships to improve clustering accuracy for unknown categories [18, 11]. GCD extends the NCD setting by allowing unlabeled samples to contain both known and unknown categories. Vaze et al. [33] proposed to learn unified feature representations via a combination of self-supervised and supervised contrastive learning, and applied a semi-supervised K-means algorithm for clustering. Cao et al. [1] proposed ORCA, a framework that explicitly addresses class distribution mismatch by introducing an uncertainty-adaptive margin to balance learning between seen and novel classes. Wen et al. [38] introduced a parametric classifier with self-distillation and entropy regularization, enabling joint optimization of representation learning and clustering. Wang et al. [36] proposed SPTNet, a two-stage adaptation framework that jointly optimizes model parameters through fine-tuning and data parameters via spatial prompt tuning, effectively aligning representations with pre-trained models. Liu et al. [19] proposed RLCD, a reciprocal learning framework with an auxiliary branch for base classification and class-wise distribution regularization, enabling mutual refinement between branches and mitigating base-class bias in generalized category discovery. Recent studies have also extended the GCD problem to more complex scenarios such as active learning [21] and federated learning [24]. Although existing GCD methods have achieved promising results, most neglect domain shifts in unlabeled data. To address this limitation, HiLo [35] tackles distributional shift by minimizing mutual information between semantic and domain features, using PatchMix [46] for augmentation and curriculum learning for training. However, its layer-wise disentanglement assumption and random patch mixing may cause noise and instability. CDAD-Net [29] further explores cross-domain category discovery by aligning source and target prototypes via entropy-driven adversarial and neighborhood-based contrastive learning, though adversarial optimization can still introduce instability.

**Unsupervised Domain Adaptation**   Current Unsupervised Domain Adaptation (UDA) approaches for cross-domain knowledge transfer are mainly categorized into two strategies. The first aims to narrow the distribution gap between source and target domains to improve generalization. Common techniques include moment matching [20, 23, 32, 7, 6] and adversarial training [30, 31, 10] to learn domain-invariant features. The second strategy focuses on employing larger and more expressive models to mitigate domain shifts. For example, TVT [41] injects transferability information into

multi-head attention modules to guide learning of transferable and domain-specific features. MTTrans [43] utilizes multi-level feature alignment and pseudo-labeling to enable cross-domain knowledge transfer. CDAC [37] enhances target domain performance through multi-consistency constraints at the attention level. In addition, recent research has explored frequency-based domain adaptation strategies [42]. These studies have found that domain-specific style information is typically encoded in the low-frequency and amplitude components of images, while semantic content is primarily captured in the phase components. By perturbing and aligning features in the frequency domain, such methods improve robustness to domain shifts. Despite these advancements, most UDA methods focus on improving performance on target domains sharing the same categories as the source and are not designed to discover new unknown categories, making them ineffective for the challenges posed by DS_GCD.

## 3   Preliminaries

**Problem Formulation**   In the DS_GCD setting, we have a labeled dataset $\mathcal{D}_l = \{x_i^l, y_i^l\}_{i=1}^{N^l}$, where each image $x_i^l \in \mathbb{R}^{H \times W \times C}$ is drawn from a known domain $\mathcal{T}_A$, and $y_i^l$ is its corresponding label. In addition, we have an unlabeled dataset $\mathcal{D}_u = \{x_i^u\}_{i=1}^{N^u}$, where images may originate from either the known domain $\mathcal{T}_A$ or an unknown domain $\mathcal{T}_B$. Note that unknown domains may also consist of multiple domains. Let $C^l$ denote the set of classes in the labeled dataset, and $C^u$ denote the set of classes in the unlabeled dataset. The labeled dataset contains only the known classes, i.e., $C^l = C^{\text{old}}$, while the unlabeled dataset contains both known and unknown classes, i.e., $C^u = C^{\text{old}} \cup C^{\text{new}}$. The number of categories in the unlabeled dataset can either be a known prior or estimated in advance using an offline class number estimation algorithm [33]. The goal of DS_GCD is to leverage the knowledge contained in the labeled dataset $\mathcal{D}_l$ to cluster the images in $\mathcal{D}_u$, regardless of whether they originate from known or unknown domains.

**Revisiting SimGCD**   SimGCD [38] is the first parametric classifier that jointly optimizes multiple objectives in an end-to-end manner. It leverages two synergistic losses: a contrastive loss and a clustering loss. Specifically, for an input image $x$, the contrastive loss consists of an unsupervised contrastive loss applied to all samples and a supervised contrastive loss applied to labeled samples, formulated as:

$$\mathcal{L}^{rep}(x) = -\frac{1}{|P(i)|} \sum_{p \in P(i)} \log \frac{\exp(z_i \cdot z_p / \tau)}{\sum_{a \in A(i)} \exp(z_i \cdot z_a / \tau)}, \tag{1}$$

where $z = G(E(x))$, $E$ and $G$ denote the feature extractor and projection head respectively. $\tau$ is the temperature parameter. $P(i)$ denotes the set of positive samples for $x_i$ in $A(i)$. For the unsupervised part, $P(i)$ contains only the augmented views of the same image. For the supervised part (on labeled data), $P(i)$ also includes samples with the same ground-truth label of $x_i$.

The clustering loss consists of two components: a cross-entropy loss on labeled samples, a self-distillation loss on all samples, which is given by:

$$\mathcal{L}^{cls}(x) = -\sum_{k=1}^{C^u} q_k(x) \log p_k(x), \tag{2}$$

where $p = F(E(x))$, where $F$ is the prototype classifier. For self-distillation loss on all samples, the supervised signal $q$ is the pseudo-label of the prediction from another view after temperature sharpening; For supervised learning on labeled samples, on the other hand, $q$ is its corresponding ground truth label. The overall optimization objective of SimGCD is thus formulated as:

$$\mathcal{L}_{SimGCD} = (1 - \beta) \sum_{x \in \mathcal{B}} (\mathcal{L}^{rep}(x) + \mathcal{L}^{cls}(x)) + \beta \sum_{x \in \mathcal{B}^l} (\mathcal{L}^{rep}(x) + \mathcal{L}^{cls}(x)) + \epsilon \triangle, \tag{3}$$

where $\triangle$ is an entropy regularization loss to prevent inactive classifier prototypes. $\beta$ and $\epsilon$ are balance hyperparameters. $\mathcal{B}$ and $\mathcal{B}^l$ represent the current batch and the labeled subset of the current batch, respectively. Although SimGCD performs well on GCD tasks within known domains, its performance drops significantly when facing unseen domain shifts due to a lack of consideration for distributional discrepancies. Next, we introduce the FREE framework, which incorporates four key innovations to effectively address domain shift in GCD tasks.

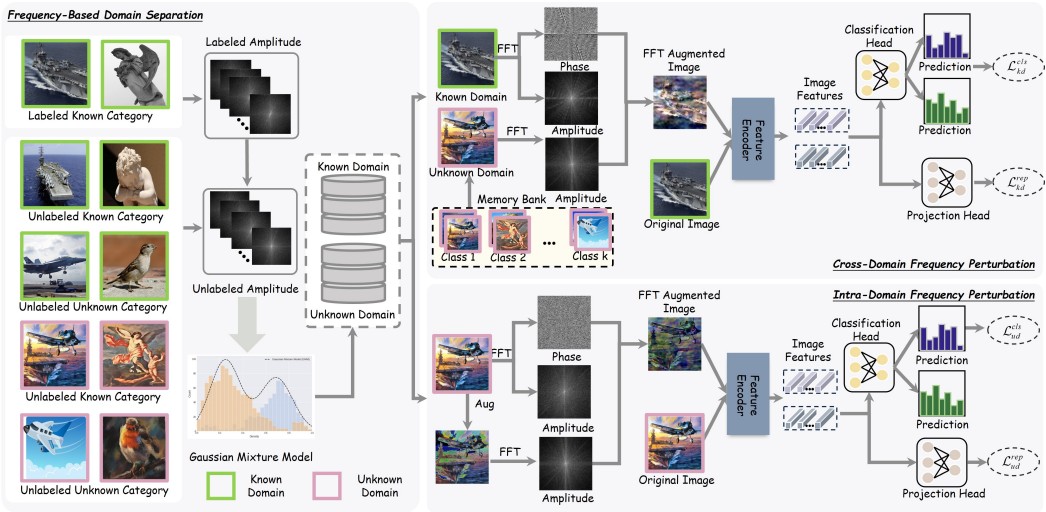

Figure 3: The overall framework of our proposed FREE method.

## 4 Frequency-guided Generalized Category Discovery Framework

The overall framework of our proposed FREE method is illustrated in Fig. 3. We first introduce a frequency-guided domain separation strategy to effectively distinguish between known and unknown domains. Then, we design domain-specific frequency perturbation strategies tailored for known and unknown domains, respectively, and jointly optimize the model using contrastive loss and clustering loss. Finally, we further propose a clustering difficulty-aware sampling strategy to enhance the model's ability to cluster challenging categories more effectively.

### 4.1 Frequency-Based Domain Separation (FDS)

Previous studies have demonstrated that the amplitude component predominantly captures the style information of an image [42, 39]. Motivated by this insight, we exploit the amplitude discrepancies between unlabeled samples and labeled anchor samples to facilitate domain separation. Specifically, given an image $x_i \in \mathbb{R}^{H \times W \times C}$, where $C$, $H$, and $W$ denote the number of channels, height, and width, respectively. We apply the Fast Fourier Transform (FFT) [4] to each channel to convert the image into the frequency domain: $\hat{x}_i(u, v) = \mathcal{F}(x_i)$. Here, $\mathcal{F}$ denotes the FFT operator. The frequency representation can then be decomposed into an amplitude component $\mathcal{A}_i$ and a phase component $\mathcal{P}_i$, defined as:

$$\mathcal{A}_i = \sqrt{\mathcal{R}(\hat{x}_i)^2(u, v) + \mathcal{I}(\hat{x}_i)^2(u, v)}, \mathcal{P}_i = \arctan\left(\frac{\mathcal{I}(\hat{x}_i)(u,v)}{\mathcal{R}(\hat{x}_i)(u,v)}\right), \tag{4}$$

where $u$ and $v$ are the corresponding frequency coordinates in the spectral domain. $\mathcal{R}(\hat{x}_i)$ and $\mathcal{I}(\hat{x}_i)$ represent the real and imaginary parts of the frequency representation $\hat{x}$. To enhance interpretability, we centralize the amplitude spectrum by shifting low-frequency components to the center of the spectrum. To determine whether an unlabeled sample originates from a known or unknown domain, we analyze the discrepancy between its amplitude component and the anchor set of amplitude representations from labeled known-domain samples. Specifically, we define a density-based function between the unlabeled amplitude feature $\mathcal{A}_i$ and the anchor set $\mathcal{D}_A$:

$$d(\mathcal{A}_i, \mathcal{D}_A) = \frac{1}{K} \sum_{\mathcal{A}_j \in \mathcal{Q}(\mathcal{A}_i, \mathcal{D}_A)} \frac{\mathcal{A}_i^\top \mathcal{A}_j}{\|\mathcal{A}_i\|_2 \cdot \|\mathcal{A}_j\|_2} \tag{5}$$

where $\mathcal{Q}(\mathcal{A}_i, \mathcal{D}_A)$ denotes the set of $K$-nearest neighbors (KNN) of the amplitude representation $\mathcal{A}_i$, retrieved from the anchor set $\mathcal{D}_A$, where each element in $\mathcal{D}_A$ corresponds to the amplitude component of a labeled sample from the known domain.

To model the distribution of this density score, we employ a Gaussian Mixture Model (GMM) [28] with two components, corresponding to known and unknown domains. Let $\zeta \in \mathcal{Z} =$

{known, unknown} be a latent variable representing domain membership. The GMM parameters are denoted as $\gamma = \{\mu_\zeta, \Sigma_\zeta, \pi_\zeta\}_{\zeta \in \mathcal{Z}}$, where $\mu_\zeta$, $\Sigma_\zeta$, and $\pi_\zeta$ denote the mean, covariance, and mixture weight of component $\zeta$, respectively. The posterior probability $p_\gamma(\zeta \mid x, \mathcal{D}_A)$ can be decomposed into: $p_\gamma(\zeta \mid x, \mathcal{D}_A) = \frac{p_\gamma(x \mid \zeta, \mathcal{D}_A) \cdot p_\gamma(\zeta \mid \mathcal{D}_A)}{p_\gamma(x \mid \mathcal{D}_A)}$, where $p_\gamma(\zeta \mid \mathcal{D}_A) = \pi_\zeta$ representing the ownership probability of $\zeta$, and $p_\gamma(x \mid \mathcal{D}_A)$ being a normalisation factor. $p_\gamma(x \mid \zeta, \mathcal{D}_A)$ is modeled using a Gaussian distribution:

$$p_\gamma(x \mid \zeta, \mathcal{D}_A) = \mathcal{N}\left(d(\mathcal{A}, \mathcal{D}_A) \mid \mu_\zeta, \Sigma_\zeta\right). \tag{6}$$

The parameters $\gamma$ are estimated via the Expectation-Maximization (EM) algorithm [3]. This probabilistic modeling enables a soft partitioning of unlabeled samples into known and unknown domains based on their amplitude similarity profiles.

## 4.2 Cross-Domain Frequency Perturbation (CDFP)

Given that the model is more familiar with the known domain and benefits from partial label supervision, training on the known domain can help the model learn domain-invariant feature representations. We propose a class-aware cross-domain frequency perturbation strategy to bridge the domain gap. Specifically, we first generate pseudo-labels for both known and unknown domain samples. Then, for each known-domain sample $x_{i,k}^{kd}$ of class $k$, we randomly select a sample $x_{j,k}^{ud}$ from the unknown domain that is also predicted as class $k$, and obtain a perturbed known-domain sample by replacing the low-frequency amplitude components:

$$\widetilde{x}_{i,k}^{kd} = \mathcal{F}^{-1}[\tilde{f}\{\mathcal{F}(x_{j,k}^{ud}), \mathcal{F}(x_{i,k}^{kd})\}]. \tag{7}$$

where $\tilde{f}$ represents the low frequency swapping operation in [42]. To mitigate the impact of noisy pseudo-labels, we perform class-aware low-frequency amplitude spectrum exchange only between samples whose predictions have confidence exceeding a predefined threshold $\eta$. To address the case where a pseudo-labeled class may be absent within a mini-batch, we maintain a memory bank $\mathcal{M}$ using a first-in-first-out (FIFO) mechanism to store the most recent $M$ unknown-domain samples. During the early training stage, when class coverage is limited, we randomly sample from the unknown domain to perform frequency perturbation. As training progresses, the memory bank gradually accumulates samples from all classes. Once the perturbed samples are generated via cross-domain frequency manipulation, we train the model using the same loss formulation as in Eq. (3), now adapted for the perturbed known-domain samples. The modified contrastive and clustering losses for cross-domain training are denoted as:

$$\mathcal{L}_{kd} = (1 - \beta) \sum_{x \in \mathcal{B}^{kd}} (\mathcal{L}_{kd}^{rep}(x) + \mathcal{L}_{kd}^{cls}(x)) + \beta \sum_{x \in \mathcal{B}^l} (\mathcal{L}_{kd}^{rep}(x) + \mathcal{L}_{kd}^{cls}(x)), \tag{8}$$

where $\mathcal{B}^{kd}$ denotes the known domain samples within the unlabeled dataset. Note that we also apply perturbation to the labeled known domain samples to fully leverage the available label information.

## 4.3 Intra-Domain Frequency Perturbation (IDFP)

To further improve the model's recognition performance on unknown domains, we propose an intra-domain frequency perturbation strategy to enhance the model's invariance to variations within the unknown domain. Specifically, for each unlabeled sample in the unknown domain, we apply two independent random data augmentations to generate two distinct views. We then apply the FFT to convert both views into the frequency domain, exchange their low-frequency amplitude components, and use the inverse Fast Fourier Transform $\mathcal{F}^{-1}$ to reconstruct the perturbed samples:

$$\widetilde{x}_i = \mathcal{F}^{-1}[\tilde{f}\{\mathcal{F}(\tilde{\mathcal{S}}(x_i)), \mathcal{F}(\mathcal{S}(x_i))\}] \tag{9}$$

where $\mathcal{S}$ and $\tilde{\mathcal{S}}$ represent different data augmentations. Note that there are two augmented samples at this stage, and we randomly select one of them for training. To encourage the model to focus on learning semantically meaningful features while remaining invariant to domain-specific styles, we employ the loss defined in Eq. (3). The key difference is that we treat the frequency-perturbed views as additional positive pairs in contrastive learning, thereby promoting robust representations. For the clustering loss, we replace the soft assignment $q$ with pseudo-labels obtained by sharpening the

predictions from the frequency-augmented counterpart. The modified contrastive and clustering loss for intra-domain learning is defined as:

$$\mathcal{L}_{ud} = \sum_{x \in \mathcal{B}^{ud}} \left( \mathcal{L}_{ud}^{rep}(x) + \mathcal{L}_{ud}^{cls}(x) \right), \tag{10}$$

where $\mathcal{B}^{ud}$ denotes the unknown domain samples within the unlabeled dataset.

Finally, by combining the losses from known-domain and unknown-domain, the overall training objective of our framework is defined as: $\mathcal{L}_{total} = \mathcal{L}_{kd} + \mathcal{L}_{ud} + \epsilon \triangle$.

### 4.4 Clustering Difficulty-Aware Sampling (CDAS)

While the above approaches have effectively alleviated the impact of domain distribution shift in GCD, it overlooks the varying levels of clustering difficulty across categories. This limitation may lead to suboptimal representation learning and clustering performance. To address this issue, we propose a clustering difficulty-aware sampling strategy, which adaptively samples data according to the difficulty level of each class. By focusing more on hard-to-cluster categories, the model can better allocate learning capacity where it is most needed. To quantify clustering difficulty, we introduce a new metric based on the class prototypes $\mathbf{O} = [o_1; \ldots; o_{C^u}]$ learned by the prototype classifier. The metric integrates two complementary components: intra-class compactness and inter-class separability. The intra-class compactness is measured by evaluating the variance of feature embeddings within each predicted class: $d_c^{\text{intra}} = \frac{1}{N_c} \sum_{\hat{y}_i = c} (E(x_i) - o_c)(E(x_i) - o_c)^\top$, where $\hat{y}_i = \arg\max_c p_i^{(c)}$ denotes the predicted label for sample $x_i$, $E(x_i)$ denotes the feature embedding of sample $x_i$, and $N_c$ is the number of samples assigned to class $c$. A higher variance $d_c^{\text{intra}}$ indicates greater dispersion of features within the class, suggesting increased clustering difficulty. The inter-class separability is designed to assess the degree of distinction between different class prototypes. Classes with closer prototype distances are more prone to confusion. This is formally defined as: $d_c^{\text{inter}} = \frac{1}{|C^u|-1} \sum_{j=1, j \neq c}^{C^u} \text{sim}(o_c, o_j)$, where $\text{sim}(\cdot, \cdot)$ is the cosine similarity, a larger $d_c^{\text{inter}}$ implies reduced inter-class margin and higher difficulty in distinguishing class $c$ from others. Based on intra-class variance and inter-class similarity, we compute a comprehensive learning difficulty score and convert it into a sampling probability for each category:

$$p_{\text{difficulty}}^c = \frac{\exp(d_c^{\text{intra}} + d_c^{\text{inter}})}{\sum_{j=1}^{C^u} \exp(d_j^{\text{intra}} + d_j^{\text{inter}})} \tag{11}$$

where $c = 1, \ldots, C^u$ is the category index. During training, a category $c$ is drawn from the categorical distribution $p_{\text{difficulty}}^c$, where $p_{\text{difficulty}}^c$ indicates the relative clustering difficulty of each class. Feature embeddings predicted by the model as belonging to the selected category are then retrieved and used in subsequent classification or contrastive learning to refine the feature space. By assigning higher sampling probabilities to more challenging categories, the model is encouraged to focus on difficult clusters, thereby improving overall clustering performance.

## 5 Experiments

### 5.1 Experimental Setup

**Data Preparation**   To validate the effectiveness of the proposed method, we conduct experiments on DomainNet [23] and the Corrupted Semantic Shift Benchmark (SSB-C) [35]. DomainNet consists of approximately 600,000 images across 345 categories, distributed over six distinct domains: Real, Clipart, Infograph, Painting, Quickdraw, and Sketch. SSB-C is built upon the Semantic Shift Benchmark (SSB), which includes three fine-grained datasets: CUB [34], Stanford Cars [15], and FGVC-Aircraft [22]. SSB-C introduces nine types of corruption to each sub-dataset to simulate domain shifts, namely: Gaussian noise, shot noise, impulse noise, zoom blur, snow, frost blur, fog, speckle noise, and spatter. Each corruption type is applied at five levels of severity, resulting in a challenging benchmark for evaluating robustness under semantic and visual distribution shifts.

Following the protocol in [35], for the DomainNet dataset, we designate the Real domain as the known domain $\mathcal{T}_A$, and treat each of the remaining domains in turn as the unknown domain $\mathcal{T}_B$ (or

alternatively, combine all remaining domains as a unified unknown domain $\mathcal{T}_B$). For the SSB-C benchmark, we follow a similar setup [35]: for each dataset, we use its original (clean) version as the known domain $\mathcal{T}_A$, and its corrupted variant as the unknown domain $\mathcal{T}_B$. Regarding category partitioning, we adopt the same practice as in [33, 35]. Specifically, for the known domain $\mathcal{T}_A$, we randomly select a subset of categories as the known classes. Then, 50% of the samples from these known classes are used to construct the labeled dataset $\mathcal{D}_l$, while the remaining samples (along with all samples from the unknown domain $\mathcal{T}_B$) form the unlabeled dataset $\mathcal{D}_u$. Detailed data splits are summarized in Table 5 in the Appendices.

**Implementation Details and Evaluation Metrics**   Following [35, 33], We used clustering accuracy as the main metric to evaluate clustering performance. For the DomainNet and SSB-C datasets, we report clustering accuracy across different subsets of the unlabeled data from the known domain $\mathcal{T}_A$ and unknown domain $\mathcal{T}_B$, including accuracy over all classes (All), Old classes only (Old), and New classes only (New). Clustering accuracy is computed as follows: $\text{ClusterAcc} = \frac{1}{N} \sum_{i=1}^{N} \mathbf{1}\{y_i = \psi(\hat{y}_i)\}$, where $y_i$ denote the ground-truth labels, $\hat{y}_i$ denote the model predictions, and $\psi$ be the optimal permutation that aligns the predicted labels with the ground-truth labels using the Hungarian algorithm [16].

Following [35, 33], we adopt the DINO [2] pretrained ViT-B/16 as our backbone network. Both the projection head and classification head are implemented as multi-layer perceptrons (MLPs). We fine-tune only the last layer of the ViT-B/16 backbone. The learning rate is initialized to 0.1 and adjusted throughout training using a cosine annealing schedule. The model is trained for 200 epochs using the SGD optimizer with a batch size of 256. The hyperparameters $\beta$, $\eta$, and $\epsilon$ are set to 0.35, 0.9, and 0.1 respectively [38], and the memory bank size $M$ is set to 1024. The number of nearest neighbors $K$ is set to 3. To accelerate K-nearest neighbor computations, we utilize the Faiss library [14]. All algorithms are run three times with different random seeds, and we report the averaged results. All models were implemented in PyTorch and trained using eight NVIDIA RTX 4090 GPUs. Please refer to the Appendices for additional training details and full hyperparameter settings of our method.

Table 1: Clustering performance of different methods on the DomainNet benchmark. For each task, we use Real as the known domain $\mathcal{T}_A$ and sequentially select one domain from Painting, Sketch, Quickdraw, Clipart, and Infograph as the unknown domain $\mathcal{T}_B$. Clustering accuracies are reported for both $\mathcal{T}_A$ and $\mathcal{T}_B$.

| Methods | Real+Painting | | | | | | Real+Sketch | | | | | | Real+Quickdraw | | | | | | Real+Clipart | | | | | | Real+Infograph | | | | | |
|---|---|---|---|---|---|---|---|---|---|---|---|---|---|---|---|---|---|---|---|---|---|---|---|---|---|---|---|---|---|---|
| | Real | | | Painting | | | Real | | | Sketch | | | Real | | | Quickdraw | | | Real | | | Clipart | | | Real | | | Infograph | | |
| | All | Old | New | All | Old | New | All | Old | New | All | Old | New | All | Old | New | All | Old | New | All | Old | New | All | Old | New | All | Old | New | All | Old | New |
| RankStats+ | 34.1 | 62.0 | 19.7 | 29.7 | 49.7 | 9.6 | 34.2 | 62.0 | 19.8 | 17.1 | 31.1 | 6.8 | 34.1 | 62.5 | 19.5 | 4.1 | 4.4 | 3.9 | 34.0 | 62.4 | 19.4 | 24.1 | 45.1 | 6.2 | 34.2 | 62.4 | 19.6 | 12.5 | 21.9 | 6.3 |
| UNO+ | 44.2 | 72.2 | 29.7 | 30.1 | 45.1 | 17.2 | 43.7 | 72.5 | 28.9 | 12.5 | 17.0 | 9.2 | 31.1 | 60.0 | 16.1 | 6.3 | 5.8 | 6.8 | 44.5 | 66.1 | 33.3 | 21.9 | 35.6 | 10.1 | 42.8 | 69.4 | 29.0 | 10.9 | 15.2 | 8.0 |
| ORCA | 31.9 | 49.8 | 23.5 | 28.7 | 38.5 | 7.1 | 32.5 | 50.0 | 23.9 | 11.4 | 14.5 | 7.2 | 19.2 | 39.1 | 15.3 | 3.4 | 3.5 | 3.2 | 32.0 | 49.7 | 23.9 | 19.1 | 31.8 | 4.3 | 29.1 | 47.7 | 20.1 | 8.6 | 13.7 | 7.1 |
| GCD | 47.3 | 53.6 | 44.1 | 32.9 | 41.8 | 23.0 | 48.0 | 53.8 | 45.3 | 16.6 | 22.4 | 11.1 | 37.6 | 41.0 | 35.2 | 5.7 | 4.2 | 6.9 | 47.7 | 53.8 | 44.3 | 22.4 | 34.4 | 16.0 | 41.9 | 46.1 | 39.0 | 10.9 | 17.1 | 8.8 |
| SimGCD | 61.3 | 77.8 | 52.9 | 34.5 | 35.6 | 33.5 | 62.4 | 77.6 | 54.6 | 16.4 | 20.2 | 13.6 | 47.4 | 64.5 | 37.4 | 6.6 | 5.8 | 7.5 | 61.6 | 77.2 | 53.6 | 23.9 | 31.5 | 17.3 | 52.7 | 67.0 | 44.8 | 11.6 | 15.4 | 9.1 |
| SPTNet | 61.6 | 76.9 | 54.7 | 35.2 | 35.9 | 35.1 | 63.3 | 77.8 | 55.3 | 16.7 | 26.0 | 11.3 | 47.1 | 65.6 | 35.4 | 6.9 | 5.7 | 7.7 | 62.5 | 76.5 | 55.4 | 24.7 | 30.9 | 18.8 | 54.5 | 67.9 | 46.2 | 11.9 | 19.4 | 7.9 |
| RLCD | 62.1 | 78.3 | 53.8 | 36.9 | 35.7 | 36.2 | 62.8 | 77.4 | 55.7 | 17.0 | 20.4 | 15.2 | 49.1 | 67.8 | 38.0 | 7.0 | 5.8 | 7.8 | 62.3 | 77.1 | 54.7 | 24.5 | 38.0 | 13.9 | 57.2 | 68.3 | 48.1 | 12.0 | 15.9 | 9.8 |
| CDAD-Net | 63.6 | 77.8 | 56.3 | 38.4 | 38.4 | 37.5 | 61.9 | 76.3 | 52.1 | 17.3 | 20.9 | 15.9 | 48.5 | 66.5 | 36.7 | 6.4 | 5.6 | 7.3 | 61.3 | 77.0 | 53.1 | 25.2 | 31.9 | 19.0 | 56.5 | 68.0 | 47.1 | 11.8 | 15.6 | 9.4 |
| HiLo | 64.4 | 77.6 | 57.5 | 42.1 | 42.9 | 41.3 | 63.3 | 77.9 | 55.9 | 19.4 | 22.4 | 17.1 | 58.6 | 76.4 | 52.5 | 7.4 | 6.9 | 8.0 | 63.8 | 77.6 | 56.6 | 27.7 | 34.6 | 21.7 | 64.2 | 78.1 | 57.0 | 13.7 | 16.4 | 11.9 |
| FREE | **67.7** | **78.1** | **61.2** | **45.6** | **46.1** | **44.8** | **67.8** | **78.2** | **61.6** | **22.5** | **25.8** | **20.9** | **61.4** | **78.1** | **55.1** | **8.9** | **7.8** | **9.0** | **66.4** | **78.1** | **60.1** | **29.3** | **37.2** | **26.3** | **68.1** | **78.9** | **60.2** | **16.1** | **18.6** | **13.4** |

Table 2: Clustering performance of different methods on the SSB-C benchmark. For each dataset (CUB, Scars, and FGVC), we use the original clean version as the known domain $\mathcal{T}_A$ and its corresponding corrupted version as the unknown domain $\mathcal{T}_B$. Clustering accuracies are reported for both $\mathcal{T}_A$ and $\mathcal{T}_B$.

| Methods | CUB-C | | | | | | Scars-C | | | | | | FGVC-C | | | | | |
|---|---|---|---|---|---|---|---|---|---|---|---|---|---|---|---|---|---|---|
| | Original | | | Corrupted | | | Original | | | Corrupted | | | Original | | | Corrupted | | |
| | All | Old | New | All | Old | New | All | Old | New | All | Old | New | All | Old | New | All | Old | New |
| RankStats+ | 19.3 | 22.0 | 15.4 | 13.6 | 23.9 | 4.5 | 14.8 | 20.8 | 7.8 | 11.5 | 22.6 | 1.0 | 14.4 | 16.4 | 14.5 | 8.3 | 15.6 | 5.0 |
| UNO+ | 25.9 | 40.1 | 21.3 | 21.5 | 33.4 | 8.6 | 22.0 | 41.8 | 7.0 | 16.9 | 29.8 | 4.5 | 22.0 | 33.4 | 15.8 | 16.5 | 25.2 | 8.8 |
| ORCA | 18.2 | 22.8 | 14.5 | 21.5 | 23.1 | 18.9 | 19.1 | 28.7 | 11.2 | 15.0 | 22.4 | 8.3 | 17.6 | 19.3 | 16.1 | 13.9 | 17.3 | 10.1 |
| GCD | 26.6 | 27.5 | 25.7 | 25.1 | 28.7 | 22.0 | 22.1 | 35.2 | 20.5 | 21.6 | 29.2 | 10.5 | 25.2 | 28.7 | 23.0 | 21.0 | 23.1 | 17.3 |
| SimGCD | 31.9 | 33.9 | 29.0 | 28.8 | 31.6 | 25.0 | 26.7 | 39.6 | 25.6 | 22.1 | 30.5 | 14.1 | 26.1 | 28.9 | 25.1 | 22.3 | 23.2 | 21.4 |
| SPTNet | 33.0 | 34.5 | 31.2 | 30.1 | 33.1 | 26.1 | 28.0 | 40.2 | 27.9 | 24.2 | 32.1 | 16.3 | 28.7 | 30.2 | 27.9 | 24.8 | 25.7 | 23.9 |
| RLCD | 35.9 | 35.1 | 33.2 | 32.3 | 34.8 | 28.5 | 29.8 | 41.2 | 30.4 | 25.3 | 33.4 | 18.1 | 27.9 | 30.1 | 26.8 | 24.4 | 26.8 | 22.7 |
| CDAD-Net | 40.4 | 38.9 | 39.3 | 37.7 | 39.1 | 34.2 | 32.1 | 42.9 | 32.2 | 28.8 | 35.6 | 21.4 | 33.8 | 35.5 | 31.2 | 27.8 | 29.6 | 25.6 |
| HiLo | 56.8 | 54.0 | 60.3 | 52.0 | 53.6 | 50.5 | 39.5 | 44.8 | 37.0 | 35.6 | 42.9 | 28.4 | 44.2 | 50.6 | 47.4 | 31.2 | 29.0 | 33.4 |
| FREE | **60.4** | **58.5** | **63.2** | **55.7** | **57.1** | **53.7** | **43.6** | **48.1** | **40.8** | **38.9** | **46.1** | **32.6** | **48.5** | **54.9** | **51.2** | **35.0** | **32.4** | **38.9** |

## 5.2 Comparison with Other SOTA Algorithms

We compare our method with several state-of-the-art approaches for GCD, including ORCA [1], GCD [33], SimGCD [38], SPTNet [36], and RLCD [19], as well as advanced methods originally designed for NCD such as RankStats+ [12] and UNO+ [9], which we adapt to fit the GCD setting. Additionally, we compare our approach against CDAD-Net [29] and HiLo [35], recent methods tailored for DS_GCD. Table 1 and Table 2 report clustering performance on the DomainNet and SSB-C datasets, respectively, with detailed per-domain results provided in Appendix. Notably, existing GCD methods suffer significant performance degradation under domain shift, and the presence of unlabeled samples from unknown domains can even impair clustering accuracy within the known domain. In contrast, our method consistently achieves performance gains across most scenarios. For instance, on the SSB-C benchmark, our method consistently outperforms the strongest baseline, HiLo. Specifically, it achieves improvements in all categories on the corrupted domains of CUB-C, Scar-C, and FGVC-C by 3.7%, 3.3%, and 3.8%, respectively. In addition to the gains on the unknown (corrupted) domains, our approach also leads to noticeable improvements in clustering accuracy within the known (clean) domains, demonstrating its ability to maintain robustness while effectively bridging domain gaps. On the more challenging DomainNet dataset, our method yields improvements in nearly all domain pairings. For example, when using Real as the known domain and Painting as the unknown domain, the proposed method outperforms HiLo by 3.5% in clustering accuracy on all categories in the Painting domain and by 3.3% in the Real domain. These results further demonstrate the robustness of our method under substantial domain discrepancy.

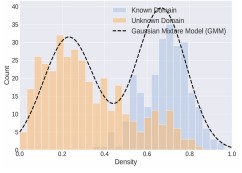
Real-Painting

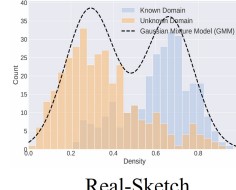
Real-Sketch

Figure 4: Density histograms for different task combinations. A two-component Gaussian mixture model is fitted to distinguish between known and unknown domain samples (domain labels are used here for visualization only).

Table 3: Ablation study of different components. The real domain from DomainNet is used as the known domain $\mathcal{T}_A$, and painting as the unknown domain $\mathcal{T}_B$. Clustering performance is reported for both domains under various component configurations.

| FDS | IDFP | CDFP | CDAS | Real All | Real Old | Real New | Painting All | Painting Old | Painting New |
|---|---|---|---|---|---|---|---|---|---|
| ✗ | ✗ | ✗ | ✗ | 61.3 | 77.8 | 52.9 | 34.5 | 35.6 | 33.5 |
| ✓ | ✓ | ✗ | ✗ | 62.0 | 77.8 | 53.3 | 36.6 | 37.8 | 35.5 |
| ✓ | ✗ | ✓ | ✗ | 65.6 | 77.9 | 58.2 | 41.6 | 41.9 | 40.7 |
| ✓ | ✗ | ✗ | ✓ | 63.1 | 77.9 | 54.8 | 37.6 | 38.0 | 36.9 |
| ✓ | ✓ | ✓ | ✗ | 66.1 | 78.0 | 59.5 | 42.8 | 43.1 | 42.0 |
| ✓ | ✓ | ✓ | ✓ | **67.7** | **78.1** | **61.2** | **45.6** | **46.1** | **44.8** |

## 5.3 Analysis and Ablation Study

**Effectiveness of different components** To better understand the contributions of each component, we conduct extensive ablation studies on the DomainNet dataset. Using SimGCD as a baseline, we incrementally add each module to observe its individual impact on model performance. As shown in Table 3, it can be observed that baseline methods that ignore domain shift struggle to achieve good performance on the unknown domain and lack robustness on the known domain. In contrast, our proposed domain separation strategy combined with domain-specific frequency perturbation effectively improves clustering performance on both known and unknown domains (Rows 2 and 3). Finally, we find that clustering difficulty-aware resampling further boosts clustering performance, particularly in the unknown domain (Row 4).

We further visualize the results of our frequency-based domain separation. As shown in Fig. 4, most samples from the known domain exhibit higher density values and appear on the right side of the histogram, while samples from the unknown domain tend to have lower density values and are located on the left. This indicates that samples with higher density are likely from the known domain, whereas those with lower density are likely from the unknown domain, thereby validating the effectiveness of our approach.

Table 4: Ablation study on different transformation strategies.

| Method | Real All | Real Old | Real New | Painting All | Painting Old | Painting New |
|---|---|---|---|---|---|---|
| Random | 62.7 | 76.5 | 54.3 | 33.2 | 34.5 | 32.5 |
| w/o class | 65.6 | 77.6 | 58.3 | 43.5 | 44.6 | 42.1 |
| FREE | **67.7** | **78.1** | **61.2** | **45.6** | **46.1** | **44.8** |

**Ablation study on different transformation strategies** We further investigate the impact of different frequency transformation strategies by comparing our proposed FREE method with random

and class-agnostic transformations. As shown in Table 4, without domain separation, simply applying random frequency perturbations fails to improve performance on both known and unknown domains, and may even lead to negative transfer (Row 1). This suggests that blindly swapping amplitude components, without accounting for the directionality of style transfer, introduces unwanted noise that hinders model learning. Moreover, we observe that class-agnostic transformations perform worse than our class-aware method (Rows 2 and 3), indicating that FREE effectively mitigates inter-class style discrepancies and further enhances clustering performance.

## 6 Conclusion and Limitations

In this paper, we propose a novel learning framework to address the GCD task under domain shift. By introducing four key innovations, our method enhances the robustness of clustering against distributional shifts. Extensive experiments on multiple domain-shifted benchmarks validate the effectiveness of the proposed approach. However, our method also has certain limitations. For instance, it primarily focuses on image classification tasks; extending this framework to more challenging scenarios such as open-set semantic segmentation and object detection remains an interesting direction for future research. Moreover, our approach requires access to both known and unknown domain data during training. An important avenue for future work is to adapt the method to domain generalization settings, where data from the unknown domain is not available during training.

## 7 Acknowledgments

This work was supported by the Center of Excellence for Antimicrobial Therapeutics Discovery and Innovation (CEATDI), Grant No. 8002003.

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

## A Technical Appendices

### A.1 Preliminaries

**Fourier-Based Interpretation of Phase and Amplitude Spectra**

Given a grayscale image $x \in \mathbb{R}^{N \times N}$, its complex-valued Fourier coefficient at frequency $(u, v)$ is defined as:

$$\mathcal{F}_x(u, v) = \sum_{n=1}^{N} \sum_{m=1}^{N} x(n, m) \cdot e^{-2\pi i(un+vm)/N} = \sum x(n, m) \cdot (\cos \theta + i \cdot \sin \theta), \quad (12)$$

where $\theta = -2\pi(un + vm)/N$.

The real and imaginary parts can thus be rewritten as:

$$\mathcal{R}_x(u, v) = \sum x(n, m) \cdot \cos \theta, \quad (13)$$

$$\mathcal{I}_x(u, v) = \sum x(n, m) \cdot \sin \theta. \quad (14)$$

Following the template-based interpretation [17], we decompose the cosine and sine components based on their signs:

$$\mathcal{T}_{\mathcal{R}^+}^{u,v}(x) = \sum_{\cos \theta \geq 0} x(n, m) \cdot \cos \theta, \qquad \mathcal{T}_{\mathcal{R}^-}^{u,v}(x) = \sum_{\cos \theta < 0} x(n, m) \cdot (-\cos \theta), \quad (15)$$

$$\mathcal{T}_{\mathcal{I}^+}^{u,v}(x) = \sum_{\sin \theta \geq 0} x(n, m) \cdot \sin \theta, \qquad \mathcal{T}_{\mathcal{I}^-}^{u,v}(x) = \sum_{\sin \theta < 0} x(n, m) \cdot (-\sin \theta). \quad (16)$$

The real and imaginary components can thus be written as:

$$\mathcal{R}_x(u, v) = \mathcal{T}_{\mathcal{R}^+}^{u,v}(x) - \mathcal{T}_{\mathcal{R}^-}^{u,v}(x), \quad (17)$$

$$\mathcal{I}_x(u, v) = \mathcal{T}_{\mathcal{I}^+}^{u,v}(x) - \mathcal{T}_{\mathcal{I}^-}^{u,v}(x). \quad (18)$$

**Phase Spectrum (Semantic / Structural Encoding)**

The phase spectrum is defined as:

$$\mathcal{P}_x(u, v) = \arctan\left(\frac{\mathcal{I}_x(u, v)}{\mathcal{R}_x(u, v)}\right) = \arctan\left(\frac{\mathcal{T}_{\mathcal{I}^+}^{u,v}(x) - \mathcal{T}_{\mathcal{I}^-}^{u,v}(x)}{\mathcal{T}_{\mathcal{R}^+}^{u,v}(x) - \mathcal{T}_{\mathcal{R}^-}^{u,v}(x)}\right). \quad (19)$$

This representation shows that the phase spectrum encodes the relative strength between different directional templates. Since it retains the positional and structural layout of features, it captures the semantic and structural information of the image, even in the absence of amplitude [17].

**Amplitude Spectrum (Style / Domain Encoding)**

The amplitude spectrum is given by:

$$\mathcal{A}_x(u,v) = \sqrt{\mathcal{R}_x(u,v)^2 + \mathcal{I}_x(u,v)^2} = \sqrt{(\mathcal{T}_{\mathcal{R}^+}^{u,v}(x) - \mathcal{T}_{\mathcal{R}^-}^{u,v}(x))^2 + (\mathcal{T}_{\mathcal{I}^+}^{u,v}(x) - \mathcal{T}_{\mathcal{I}^-}^{u,v}(x))^2}. \quad (20)$$

This expression shows that the amplitude spectrum measures the absolute energy of the frequency response. It is sensitive to image-level traits such as texture, contrast, and lighting, and is thus considered to capture the domain style or appearance of the image.

**Summary.** The phase spectrum encodes structural and semantic information via the relative template responses, while the amplitude spectrum encodes domain-specific appearance through energy distributions. This decomposition provides a theoretical basis for frequency-based domain manipulation and adaptation strategies.

Table 5: Detailed statistics of data split.

| Dataset | Labelled | | | Unlabelled | | |
|---|---|---|---|---|---|---|
| | #Image | #Class | #Domain | #Image | #Class | #Domain |
| DomainNet | 39.1K | 172 | 1 | 547.5K | 345 | 6 |
| CUB-C | 1.5K | 100 | 1 | 45K | 200 | 10 |
| Scars-C | 2.0K | 98 | 1 | 61K | 196 | 10 |
| FGVC-C | 1.7K | 50 | 1 | 50K | 100 | 10 |

# B  More Empirical Results

## B.1  Detailed Description of the Dataset

**DomainNet.** DomainNet [23] is one of the largest and most challenging benchmark datasets for domain adaptation and generalization tasks. It consists of approximately 600,000 images covering 345 object categories across six distinct domains, each representing a different visual style or modality. The domains are:

- **Real**: Natural images collected from online sources (e.g., photos).
- **Clipart**: Cartoon-style clipart graphics.
- **Sketch**: Hand-drawn sketches.
- **Painting**: Artistic paintings with various styles.
- **Infograph**: Informative icons and charts, often stylized and abstract.
- **Quickdraw**: Simplified sketches collected via the Google Quick, Draw! project.

**SSB-C Dataset.** SSB-C [35] is a perturbed version of the Semantic Shift Benchmark (SSB). SSB consists of three fine-grained visual recognition datasets:

- **CUB-200-2011 (CUB):** A bird species classification dataset with 200 categories.
- **Stanford Cars (SCAR):** A fine-grained car model classification dataset containing 196 categories.
- **FGVC Aircraft (FGVC):** An aircraft model classification dataset with 100 categories.

In SSB-C, each of these datasets is augmented with **nine types of perturbations**, each applied at **five severity levels**, resulting in a dataset that is **45 times larger** than the original SSB.

Table 5 presents the detailed splits of the different datasets.

## B.2 Multiple unseen domains for Domainnet

We further explored the scenario where the unknown domain consists of multiple domains to more comprehensively validate the effectiveness of the proposed method. Specifically, we combined the five domains in DomainNet—excluding the real domain—into a single aggregated unknown domain $\mathcal{T}_B$, and then re-ran all the experiments under this setting. As shown in Table 6, even when the unknown domain is composed of multiple diverse subdomains, our proposed FREE framework consistently outperforms all competing baseline methods. This result not only demonstrates the robustness of our approach in more complex, multi-domain unknown settings but also highlights its strong generalization capability across varied domain shifts.

Table 6: Clustering performance when $\mathcal{T}_B$ contains multiple unknown domains. We construct $\mathcal{T}_B$ by combining the five domains from DomainNet excluding the Real domain, and report clustering accuracy separately for each domain.

| Methods | Real | | | Painting | | | Sketch | | | Quickdraw | | | Clipart | | | Infograph | | |
|---|---|---|---|---|---|---|---|---|---|---|---|---|---|---|---|---|---|---|
| | All | Old | New | All | Old | New | All | Old | New | All | Old | New | All | Old | New | All | Old | New |
| RankStats+ | 34.0 | 62.3 | 19.9 | 30.3 | 50.1 | 11.1 | 17.9 | **31.5** | 7.2 | 2.4 | 2.0 | 2.5 | 25.1 | **46.4** | 6.3 | 12.0 | **22.1** | 5.5 |
| UNO+ | 43.1 | 72.0 | 28.6 | 30.3 | 43.7 | 17.4 | 12.0 | 16.3 | 8.9 | 2.1 | 2.3 | 1.8 | 22.8 | 37.4 | 9.5 | 12.4 | 20.3 | 6.5 |
| ORCA | 32.1 | 49.9 | 23.5 | 23.0 | 38.8 | 17.0 | 11.6 | 14.7 | 7.6 | 2.8 | 3.6 | 2.1 | 20.1 | 33.4 | 10.3 | 8.4 | 17.8 | 6.8 |
| GCD | 47.8 | 53.5 | 45.1 | 32.9 | 40.3 | 26.9 | 17.0 | 22.7 | 11.3 | 1.9 | 2.4 | 1.8 | 24.3 | 31.2 | 15.1 | 10.5 | 12.0 | 9.9 |
| SimGCD | 62.2 | 77.3 | 54.3 | 36.6 | 42.9 | 30.3 | 18.2 | 22.6 | 15.0 | 2.2 | 2.0 | 2.4 | 25.0 | 34.7 | 16.4 | 11.8 | 13.8 | 10.5 |
| SPTNet | 63.1 | 77.4 | 55.5 | 38.4 | 45.1 | 32.5 | 19.3 | 23.3 | 16.7 | 2.4 | 2.7 | 2.5 | 25.3 | 34.1 | 15.9 | 12.3 | 15.1 | 11.7 |
| RLCD | 64.3 | 77.6 | 57.0 | 39.7 | 46.2 | 33.8 | 20.5 | 27.5 | 15.5 | 2.6 | 2.5 | 3.0 | 26.7 | 35.9 | 18.3 | 14.5 | 16.4 | 13.0 |
| CDAD-Net | 64.6 | 76.5 | 59.5 | 40.1 | 45.6 | 36.4 | 18.6 | 22.1 | 16.0 | 1.9 | 2.6 | 1.3 | 26.3 | 33.2 | 19.9 | 12.7 | 15.0 | 12.8 |
| HiLo | 65.8 | 77.8 | 58.9 | 43.4 | 49.0 | 42.9 | 20.0 | 23.6 | 17.4 | 3.1 | 4.0 | 2.5 | 27.6 | 34.7 | 21.4 | 13.9 | 16.5 | 12.1 |
| FREE | **67.4** | **78.5** | **61.3** | **48.9** | **54.6** | **48.6** | **22.7** | 25.1 | **19.1** | **4.5** | **5.2** | **3.1** | **29.7** | 35.4 | **25.7** | **16.1** | 20.1 | **14.5** |

## B.3 Additional experimental results on SSB-C

We additionally report the performance of our method on each corrupted dataset in the SSB-C benchmark. As shown in Table 7, Table 8 and Table 9, our proposed approach consistently outperforms baseline methods on most of the corrupted datasets, demonstrating its robustness and effectiveness across various challenging scenarios.

Table 7: Detailed clustering performance on CUB-C. We report the clustering accuracy on each corrupted domain.

| Methods | Gaussian Noise | | | Shot Noise | | | Impulse Noise | | | Zoom Blur | | | Snow | | | Frost | | | Fog | | | Speckle | | | Spatter | | |
|---|---|---|---|---|---|---|---|---|---|---|---|---|---|---|---|---|---|---|---|---|---|---|---|---|---|---|---|
| | All | Old | New | All | Old | New | All | Old | New | All | Old | New | All | Old | New | All | Old | New | All | Old | New | All | Old | New | All | Old | New |
| RankStats+ | 13.6 | 20.9 | 4.5 | 12.7 | 28.4 | 5.1 | 12.3 | 27.4 | 5.4 | 15.2 | 33.7 | 4.9 | 16.0 | 34.7 | 5.6 | 17.5 | 38.4 | 4.8 | 18.7 | 40.7 | 4.9 | 16.8 | 36.5 | 5.3 | 22.3 | 48.1 | 4.7 |
| UNO+ | 18.5 | 32.4 | 7.6 | 17.2 | 30.5 | 7.2 | 17.1 | 31.1 | 6.2 | 20.4 | 35.7 | 8.4 | 20.7 | 35.6 | 7.0 | 20.7 | 35.2 | 7.4 | 30.2 | 52.2 | 10.5 | 22.9 | 42.0 | 8.4 | 29.7 | 52.7 | 11.2 |
| ORCA | 21.5 | 23.1 | 19.9 | 21.2 | 23.7 | 18.8 | 21.1 | 23.1 | 19.2 | 20.4 | 22.0 | 18.9 | 20.1 | 22.1 | 18.3 | 22.0 | 25.5 | 18.5 | 19.2 | 20.4 | 18.0 | 22.4 | 20.8 | 19.1 | 24.8 | 31.3 | 18.3 |
| GCD | 23.4 | 22.7 | 20.0 | 22.7 | 20.4 | 31.0 | 21.9 | 20.3 | 19.6 | 25.1 | 25.3 | 21.0 | 23.2 | 22.9 | 20.2 | 23.9 | 23.1 | 20.8 | 29.7 | 31.1 | 24.4 | 27.6 | 26.7 | 24.6 | 35.2 | 36.2 | 30.3 |
| SimGCD | 23.8 | 26.6 | 22.0 | 21.6 | 23.8 | 20.4 | 20.4 | 22.5 | 19.4 | 30.5 | 35.8 | 26.2 | 29.0 | 34.3 | 24.9 | 29.1 | 32.6 | 26.7 | 33.0 | 36.9 | 30.1 | 27.3 | 29.6 | 26.1 | 41.5 | 47.0 | 37.0 |
| SPTNet | 25.5 | 28.3 | 23.8 | 23.2 | 25.7 | 22.2 | 22.4 | 24.3 | 21.2 | 32.4 | 37.6 | 28.0 | 30.9 | 36.4 | 26.6 | 30.7 | 34.6 | 28.4 | 34.8 | 38.7 | 31.9 | 28.9 | 31.4 | 27.7 | 43.2 | 48.9 | 38.6 |
| RLCD | 26.5 | 29.4 | 24.9 | 26.4 | 26.8 | 23.2 | 23.4 | 25.5 | 22.6 | 33.3 | 39.0 | 29.4 | 32.0 | 37.1 | 28.1 | 31.2 | 35.3 | 29.1 | 35.5 | 39.7 | 33.0 | 29.5 | 32.2 | 28.6 | 44.4 | 50.1 | 39.9 |
| CDAD-Net | 31.9 | 35.2 | 29.6 | 30.5 | 33.1 | 28.4 | 28.2 | 30.4 | 26.8 | 38.3 | 44.0 | 33.5 | 37.6 | 42.5 | 34.1 | 36.9 | 41.0 | 34.4 | 39.7 | 43.9 | 37.4 | 34.5 | 37.2 | 33.0 | 49.7 | **55.6** | 44.6 |
| HiLo | 41.8 | 39.8 | 43.9 | 41.0 | 38.7 | 43.3 | 42.2 | 39.8 | 44.5 | 47.9 | 43.9 | 51.8 | 49.3 | 45.8 | 52.8 | 48.5 | 45.5 | 51.4 | 50.6 | 46.8 | 54.3 | 47.9 | 45.4 | 50.2 | 50.9 | 47.2 | 54.7 |
| FREE | **45.7** | **44.9** | **48.2** | **46.5** | **43.2** | **47.2** | **47.1** | **43.5** | **47.8** | **50.3** | **48.1** | **54.2** | **53.4** | **49.7** | **55.9** | **51.2** | **48.7** | **54.4** | **53.4** | **49.8** | **57.1** | **50.3** | **48.7** | **53.1** | **53.8** | 51.2 | **56.9** |

Table 8: Detailed clustering performance on Scars-C. We report the clustering accuracy on each corrupted domain.

| Methods | Gaussian Noise | | | Shot Noise | | | Impulse Noise | | | Zoom Blur | | | Snow | | | Frost | | | Fog | | | Speckle | | | Spatter | | |
|---|---|---|---|---|---|---|---|---|---|---|---|---|---|---|---|---|---|---|---|---|---|---|---|---|---|---|---|
| | All | Old | New | All | Old | New | All | Old | New | All | Old | New | All | Old | New | All | Old | New | All | Old | New | All | Old | New | All | Old | New |
| RankStats+ | 8.5 | 16.6 | 1.6 | 8.9 | 16.7 | 1.7 | 7.2 | 13.8 | 1.5 | 11.7 | 22.9 | 0.5 | 8.9 | 17.0 | 1.3 | 11.4 | 21.9 | 0.7 | 16.8 | 32.6 | 1.2 | 12.7 | 24.1 | 1.6 | 17.3 | 34.1 | 1.6 |
| UNO+ | 13.9 | 24.8 | 6.5 | 14.0 | 25.0 | 6.9 | 11.2 | 20.4 | 6.4 | 17.1 | 33.2 | 2.6 | 13.3 | 24.0 | 4.5 | 17.3 | 29.9 | 6.3 | 22.4 | 39.8 | 3.8 | 18.6 | 33.1 | 7.1 | 21.8 | 38.4 | 4.0 |
| ORCA | 12.0 | 31.4 | 9.3 | 13.2 | 31.8 | 9.7 | 11.8 | 29.2 | 9.2 | 14.5 | 38.2 | 7.9 | 12.5 | 32.6 | 9.5 | 15.7 | 36.4 | 10.0 | 20.3 | 47.7 | 5.8 | 17.0 | 39.4 | 10.5 | 21.6 | 48.8 | 10.6 |
| GCD | 17.6 | 24.2 | 10.8 | 17.1 | 24.6 | 11.2 | 14.4 | 20.9 | 11.0 | 23.2 | 31.8 | 8.0 | 18.5 | 25.5 | 8.4 | 23.2 | 31.1 | 10.2 | 27.1 | 40.8 | 5.7 | 22.6 | 30.1 | 12.4 | 31.0 | 43.1 | 7.1 |
| SimGCD | 18.1 | 23.5 | 15.7 | 18.3 | 23.5 | 15.5 | 15.2 | 19.0 | 15.4 | 24.4 | 32.7 | 13.1 | 19.7 | 26.4 | 12.9 | 23.9 | 31.9 | 13.3 | 28.0 | 38.6 | 12.7 | 23.4 | 30.6 | 16.4 | 32.4 | 45.4 | 13.1 |
| SPTNet | 19.8 | 25.3 | 17.5 | 20.3 | 25.1 | 17.1 | 17.1 | 20.8 | 17.2 | 26.0 | 34.8 | 14.9 | 21.5 | 28.3 | 14.6 | 25.7 | 33.9 | 14.9 | 30.0 | 40.2 | 14.8 | 25.3 | 32.7 | 18.3 | 34.2 | 47.6 | 15.1 |
| RLCD | 20.9 | 26.3 | 18.7 | 21.4 | 26.8 | 18.5 | 17.8 | 21.7 | 18.4 | 27.0 | 35.9 | 16.2 | 22.9 | 29.5 | 15.8 | 27.3 | 35.1 | 15.9 | 31.2 | 42.0 | 15.7 | 26.4 | 33.6 | 19.2 | 35.6 | 49.0 | 16.3 |
| CDAD-Net | 22.6 | 28.3 | 20.3 | 23.2 | 29.0 | 20.0 | 19.3 | 24.0 | 20.1 | 28.7 | 38.0 | 17.7 | 24.4 | 31.4 | 17.2 | 29.1 | 37.4 | 17.4 | 33.5 | 45.3 | 17.1 | 28.1 | 35.5 | 20.6 | 37.4 | 49.0 | 18.1 |
| HiLo | 31.0 | 38.0 | 24.3 | 31.5 | 38.3 | 24.9 | 30.2 | 36.6 | 23.9 | 38.4 | 45.1 | 31.9 | 36.8 | 44.9 | 29.0 | 36.5 | 43.8 | 29.5 | 40.7 | 49.5 | 32.2 | 37.1 | 37.1 | 29.6 | 37.9 | 45.4 | 30.6 |
| FREE | **35.6** | **42.8** | **27.9** | **35.4** | **42.0** | **27.8** | **34.9** | **41.9** | **27.6** | **42.2** | **50.2** | **35.8** | **41.1** | **49.6** | **34.5** | **41.2** | **47.2** | **33.5** | **45.4** | **53.1** | **35.9** | **41.2** | 41.3 | **33.7** | **42.1** | **49.9** | **34.8** |

## B.4 Additional experimental results on domainnet

We further conducted additional experiments on the DomainNet dataset to comprehensively evaluate the generalization ability of our proposed method. Specifically, we first selected Real as the known

Table 9: Detailed clustering performance on FGVC-C. We report the clustering accuracy on each corrupted domain.

| Methods | Gaussian Noise | | | Shot Noise | | | Impulse Noise | | | Zoom Blur | | | Snow | | | Frost | | | Fog | | | Speckle | | | Spatter | | |
|---|---|---|---|---|---|---|---|---|---|---|---|---|---|---|---|---|---|---|---|---|---|---|---|---|---|---|---|
| | All | Old | New | All | Old | New | All | Old | New | All | Old | New | All | Old | New | All | Old | New | All | Old | New | All | Old | New | All | Old | New |
| RankStats+ | 7.3 | 13.6 | 5.0 | 6.3 | 10.7 | 5.8 | 6.0 | 10.7 | 5.3 | 10.1 | 19.9 | 4.3 | 6.2 | 12.5 | 3.8 | 8.9 | 17.7 | 4.1 | 12.5 | 24.4 | 4.5 | 7.6 | 14.0 | 5.2 | 10.5 | 20.1 | 4.9 |
| UNO+ | 15.5 | 25.2 | 5.8 | 13.5 | 22.1 | 4.9 | 13.2 | 20.1 | 6.2 | 20.1 | 28.2 | 12.0 | 15.6 | 21.3 | 9.9 | 17.6 | 25.2 | 9.9 | 19.3 | 26.9 | 11.7 | 16.5 | 27.2 | 5.6 | 20.9 | 29.6 | 12.3 |
| ORCA | 11.9 | 17.3 | 11.1 | 11.1 | 15.6 | 11.3 | 10.9 | 15.9 | 10.3 | 15.2 | 24.3 | 8.5 | 11.3 | 15.4 | 9.1 | 12.6 | 22.1 | 9.3 | 16.7 | 28.9 | 9.1 | 12.2 | 18.8 | 10.4 | 15.0 | 25.1 | 9.3 |
| GCD | 16.0 | 20.1 | 14.3 | 13.8 | 19.1 | 11.5 | 12.3 | 16.0 | 13.4 | 27.7 | 25.4 | 24.1 | 19.1 | 17.7 | 15.2 | 23.9 | 24.0 | 18.2 | 31.8 | 30.1 | 24.7 | 16.1 | 27.0 | 14.9 | 28.7 | 30.7 | 25.9 |
| SimGCD | 16.3 | 16.2 | 18.4 | 14.2 | 14.5 | 16.0 | 13.7 | 13.0 | 16.5 | 28.9 | 31.4 | 28.4 | 20.0 | 22.4 | 19.5 | 24.5 | 29.2 | 21.9 | 31.9 | **37.8** | 28.0 | 16.8 | 18.0 | 17.7 | 29.8 | 32.9 | 28.6 |
| SPTNet | 17.6 | 17.5 | 19.9 | 15.6 | 15.7 | 17.7 | 14.9 | 14.5 | 18.2 | 30.4 | 33.2 | 30.0 | 21.8 | 24.1 | 21.3 | 25.7 | 30.5 | 23.7 | 33.2 | 36.1 | 29.3 | 18.3 | 19.4 | 19.0 | 31.0 | 34.5 | 30.3 |
| RLCD | 18.7 | 18.6 | 20.9 | 16.8 | 17.2 | 18.9 | 15.9 | 15.5 | 19.4 | 31.4 | 34.3 | 31.1 | 22.4 | 25.0 | 22.3 | 27.1 | 31.5 | 25.0 | 34.3 | 36.3 | 30.5 | 19.6 | 20.8 | 20.2 | 32.1 | 35.1 | 31.4 |
| CDAD-Net | 21.3 | 21.5 | 23.8 | 19.6 | 20.0 | 21.4 | 18.6 | 18.5 | 22.0 | 34.5 | **37.2** | 34.3 | 25.5 | 27.6 | 26.0 | 30.1 | 32.7 | 29.4 | 31.7 | 36.0 | 27.3 | 22.4 | 24.0 | 23.5 | 34.8 | | 31.3 |
| HiLo | 28.6 | 25.2 | 32.0 | 26.8 | 24.4 | 29.2 | 27.9 | 24.5 | 31.4 | 36.8 | 34.2 | 39.4 | 27.8 | 27.9 | 27.8 | 33.4 | 30.4 | 36.4 | 35.8 | 34.1 | 37.5 | 30.4 | 30.4 | 32.7 | 33.4 | 32.4 | 34.4 |
| FREE | **33.7** | **28.4** | **34.5** | **31.2** | **27.2** | **32.5** | **32.0** | **27.9** | **34.6** | **39.9** | 36.3 | **41.3** | **30.4** | **30.5** | **30.9** | **36.1** | **33.1** | **38.7** | **37.9** | 37.6 | **40.9** | **34.5** | **34.2** | **36.8** | **37.3** | **35.6** | **37.9** |

domain $\mathcal{T}_A$, and then, one by one, selected each of the remaining five domains as the target domain $\mathcal{T}_B$. For each such pair, we evaluated the model's clustering performance on the four remaining domains not involved in training. As shown in Table 10, Table 11, Table 12, Table 13, Table 14, Table 15, Table 16, Table 17, our method consistently achieves performance gains in most scenarios and outperforms other competing approaches. These results demonstrate the effectiveness of our method in learning domain-invariant semantic representations, enabling better generalization across diverse visual domains.

Table 10: We use Real as the known domain ($\mathcal{T}_A$) and Sketch as the unknown domain ($\mathcal{T}_B$), and report clustering performance on $\mathcal{T}_A$, $\mathcal{T}_B$, as well as all other domains.

| Methods | Real | | | Sketch | | | others | | |
|---|---|---|---|---|---|---|---|---|---|
| | All | Old | New | All | Old | New | All | Old | New |
| RankStats+ | 34.2 | 62.0 | 19.8 | 17.1 | **31.1** | 6.8 | 17.3 | **30.0** | 6.1 |
| UNO+ | 43.7 | 72.5 | 28.9 | 12.5 | 17.0 | 9.2 | 17.4 | 26.4 | 9.5 |
| ORCA | 32.5 | 50.0 | 23.9 | 11.4 | 14.5 | 7.2 | 13.3 | 23.1 | 9.1 |
| GCD | 48.0 | 53.8 | 45.3 | 16.6 | 22.4 | 11.1 | 20.7 | 25.8 | 15.8 |
| SimGCD | 62.4 | 77.6 | 54.6 | 16.4 | 20.2 | 13.6 | 20.4 | 25.4 | 16.1 |
| SPTNet | 62.7 | 77.8 | 54.9 | 16.9 | 20.8 | 13.9 | 21.2 | 25.9 | 16.9 |
| RLCD | 63.0 | 77.6 | 55.6 | 17.4 | 20.3 | 15.6 | 21.4 | 26.4 | 16.7 |
| CDAD-Net | 62.5 | 77.4 | 55.1 | 16.6 | 20.2 | 14.1 | 22.1 | 27.1 | 16.9 |
| HiLo | 63.3 | 77.9 | 55.9 | 19.4 | 22.4 | 17.1 | 21.3 | 25.8 | 17.4 |
| FREE | **65.8** | **78.4** | **57.2** | **22.5** | 25.1 | **19.8** | **24.0** | 27.0 | **19.8** |

Table 11: Detailed clustering performance on other domains when using Real as $\mathcal{T}_A$ and Sketch as $\mathcal{T}_B$.

| Methods | Painting | | | Quickdraw | | | Clipart | | | Infograph | | |
|---|---|---|---|---|---|---|---|---|---|---|---|---|
| | All | Old | New | All | Old | New | All | Old | New | All | Old | New |
| RankStats+ | 29.7 | **49.2** | 10.2 | 2.3 | 2.1 | 2.4 | 24.6 | **45.9** | 5.9 | 12.5 | 22.6 | 5.9 |
| UNO+ | 30.8 | 44.0 | 17.6 | 2.4 | 2.4 | 2.3 | 23.1 | 38.0 | 10.1 | 13.2 | 21.2 | 7.9 |
| ORCA | 23.1 | 39.1 | 17.2 | 2.5 | **3.0** | 2.0 | 19.7 | 33.1 | 10.0 | 8.9 | 18.1 | 7.0 |
| GCD | 32.6 | 40.1 | 31.5 | 1.6 | 1.9 | 1.5 | 24.1 | 31.1 | 14.9 | 14.1 | 16.2 | 10.2 |
| SimGCD | 38.7 | 44.7 | 32.7 | 1.9 | 1.2 | **2.5** | 25.2 | 35.3 | 16.3 | 15.8 | 20.3 | 12.8 |
| SPTNet | 37.9 | 44.2 | 32.2 | 2.0 | 1.5 | 2.1 | 25.7 | 35.8 | 16.9 | 16.2 | 20.9 | 13.1 |
| RLCD | 39.0 | 44.4 | 33.1 | 1.6 | 1.1 | 2.2 | 26.2 | 35.8 | 17.2 | 16.2 | 20.8 | 13.2 |
| CDAD-Net | 38.0 | 44.1 | 32.8 | 2.0 | 1.3 | 2.3 | 25.8 | 36.1 | 16.7 | 17.1 | 21.3 | 14.8 |
| HiLo | 39.8 | 44.7 | 34.9 | 1.9 | 2.0 | 1.7 | 27.2 | 35.9 | 19.6 | 16.2 | 20.5 | 13.4 |
| FREE | **41.9** | 45.8 | **37.2** | **2.5** | 2.9 | 2.3 | **29.8** | 36.5 | **21.8** | **18.1** | **22.8** | **15.5** |

Table 12: We use Real as the known domain ($\mathcal{T}_A$) and Quickdraw as the unknown domain ($\mathcal{T}_B$), and report clustering performance on $\mathcal{T}_A$, $\mathcal{T}_B$, as well as all other domains.

| Methods | Real | | | Quickdraw | | | others | | |
|---|---|---|---|---|---|---|---|---|---|
| | All | Old | New | All | Old | New | All | Old | New |
| RankStats+ | 34.1 | 62.5 | 19.5 | 4.1 | 4.4 | 3.9 | 21.0 | **37.4** | 7.2 |
| UNO+ | 31.1 | 60.0 | 16.1 | 6.3 | 5.8 | 6.8 | 18.6 | 32.2 | 7.0 |
| ORCA | 19.2 | 39.1 | 15.3 | 3.4 | 3.5 | 3.2 | 15.6 | 28.4 | 8.1 |
| GCD | 37.6 | 41.0 | 35.2 | 5.7 | 4.2 | 6.9 | 21.9 | 34.3 | 12.2 |
| SimGCD | 47.4 | 64.5 | 37.4 | 6.6 | 5.8 | 7.5 | 22.9 | 33.8 | 13.8 |
| SPTNet | 47.8 | 64.9 | 37.6 | 6.8 | 5.9 | 7.8 | 23.1 | 33.6 | 14.5 |
| RLCD | 49.2 | 67.1 | 38.2 | 6.9 | 5.6 | 8.5 | 25.1 | 34.3 | 15.1 |
| CDAD-Net | 51.3 | 66.7 | 49.4 | 7.1 | 6.2 | 7.9 | 25.3 | 35.8 | 15.9 |
| HiLo | 58.6 | 76.4 | 52.5 | 7.4 | 6.9 | 8.0 | 25.9 | 32.5 | 20.4 |
| FREE | **62.3** | **78.9** | **56.3** | **8.1** | **8.1** | **8.9** | **27.6** | 33.9 | **22.1** |

Table 13: Detailed clustering performance on other domains when using Real as $\mathcal{T}_A$ and Quickdraw as $\mathcal{T}_B$.

| Methods | Painting | | | Sketch | | | Clipart | | | Infograph | | |
|---|---|---|---|---|---|---|---|---|---|---|---|---|
| | All | Old | New | All | Old | New | All | Old | New | All | Old | New |
| RankStats+ | 29.6 | **49.0** | 10.2 | 17.1 | **32.1** | 6.1 | 24.8 | **45.4** | 6.7 | 12.6 | **23.1** | 5.7 |
| UNO+ | 26.8 | 43.7 | 9.9 | 14.7 | 25.6 | 6.6 | 20.7 | 38.4 | 5.1 | 12.2 | 21.0 | 6.4 |
| ORCA | 22.2 | 40.9 | 10.1 | 11.9 | 22.4 | 7.1 | 17.5 | 35.6 | 5.7 | 10.3 | 18.7 | 6.6 |
| GCD | 32.9 | 45.7 | 21.4 | 18.5 | 30.5 | 10.8 | 23.5 | 39.0 | 10.7 | 13.8 | 22.1 | 7.6 |
| SimGCD | 33.8 | 45.1 | 22.5 | 19.4 | 30.1 | 11.5 | 24.0 | 38.5 | 11.4 | 14.5 | 21.6 | 9.8 |
| SPTNet | 34.5 | 46.1 | 24.2 | 21.3 | 32.1 | 11.7 | 26.2 | 39.5 | 11.8 | 15.6 | 23.1 | 9.9 |
| RLCD | 36.2 | 46.0 | 25.1 | 21.8 | 31.8 | 12.5 | 26.0 | 39.1 | 11.7 | 16.5 | 22.4 | 10.9 |
| CDAD-Net | 36.8 | 46.9 | 24.3 | 21.9 | 30.8 | 14.6 | 26.8 | 38.7 | 14.2 | 15.7 | 22.3 | 11.8 |
| HiLo | 38.6 | 45.1 | 32.2 | 22.9 | 28.8 | 18.5 | 26.0 | 36.4 | 16.9 | 16.2 | 19.8 | 13.9 |
| FREE | **42.1** | 46.5 | **36.5** | **25.1** | 31.9 | **20.1** | **29.2** | 38.9 | **20.7** | **19.1** | 21.5 | **15.7** |

Table 14: We use Real as the known domain ($\mathcal{T}_A$) and Clipart as the unknown domain ($\mathcal{T}_B$), and report clustering performance on $\mathcal{T}_A$, $\mathcal{T}_B$, as well as all other domains.

| Methods | Real | | | Clipart | | | others | | |
|---|---|---|---|---|---|---|---|---|---|
| | All | Old | New | All | Old | New | All | Old | New |
| RankStats+ | 34.0 | 62.4 | 19.4 | 24.1 | **45.1** | 6.2 | 15.8 | **27.0** | 6.4 |
| UNO+ | 44.5 | 66.1 | 33.3 | 21.9 | 35.6 | 10.1 | 16.2 | 23.2 | 10.5 |
| ORCA | 32.0 | 49.7 | 23.9 | 19.1 | 31.8 | 4.3 | 13.7 | 19.9 | 8.6 |
| GCD | 47.7 | 53.8 | 44.3 | 22.4 | 34.4 | 16.0 | 18.0 | 24.1 | 12.1 |
| SimGCD | 61.6 | 77.2 | 53.6 | 23.9 | 31.5 | 17.3 | 19.2 | 23.6 | 15.6 |
| SPTNet | 63.0 | 77.7 | 53.9 | 24.4 | 31.8 | 17.9 | 21.2 | 24.5 | 16.7 |
| RLCD | 63.1 | 77.8 | 54.1 | 24.9 | 32.3 | 18.5 | 22.0 | 25.1 | 17.0 |
| CDAD-Net | 62.9 | 77.6 | 53.8 | 25.8 | 33.0 | 18.1 | 22.2 | 25.7 | 16.1 |
| HiLo | 63.8 | 77.6 | 56.6 | 27.7 | 34.6 | 21.7 | 19.8 | 23.6 | 16.8 |
| FREE | **66.1** | **78.3** | **60.1** | **29.4** | 37.1 | **24.9** | **23.4** | 23.9 | **20.1** |

Table 15: Detailed clustering performance on other domains when using Real as $\mathcal{T}_A$ and Clipart as $\mathcal{T}_B$.

| Methods | Painting | | | Quickdraw | | | Sketch | | | Infograph | | |
|---|---|---|---|---|---|---|---|---|---|---|---|---|
| | All | Old | New | All | Old | New | All | Old | New | All | Old | New |
| RankStats+ | 30.0 | **50.3** | 9.7 | 2.6 | 2.3 | 2.9 | 17.4 | **31.9** | 6.8 | 13.1 | **23.6** | 6.2 |
| UNO+ | 31.5 | 43.3 | 19.6 | 2.8 | 2.1 | **3.6** | 17.3 | 26.8 | 10.2 | 13.3 | 20.6 | 8.5 |
| ORCA | 29.3 | 36.9 | 9.2 | 1.3 | 1.5 | 1.2 | 13.7 | 21.9 | 8.3 | 10.3 | 19.4 | 6.3 |
| GCD | 33.4 | 40.4 | 22.2 | **3.6** | **5.7** | 2.2 | 19.5 | 27.7 | 12.7 | 15.5 | 22.7 | 11.1 |
| SimGCD | 39.0 | 45.9 | 32.1 | 0.8 | 0.5 | 1.1 | 21.1 | 27.3 | 16.5 | 15.9 | 20.8 | 12.7 |
| SPTNet | 40.2 | 46.4 | 33.1 | 0.6 | 0.4 | 1.0 | 22.3 | 27.9 | 17.1 | 16.1 | 20.1 | **13.5** |
| RLCD | 41.7 | 47.4 | 34.7 | 1.2 | 0.9 | 1.3 | **23.5** | 28.8 | 18.5 | 16.2 | 20.4 | 13.1 |
| CDAD-Net | 41.0 | 46.8 | 33.8 | 1.0 | 0.8 | 1.1 | 22.6 | 27.0 | 17.2 | 15.8 | 20.1 | 11.2 |
| HiLo | 40.7 | 46.3 | 35.1 | 1.3 | 0.4 | 2.3 | 21.2 | 26.9 | 17.0 | 15.9 | 20.6 | 12.8 |
| FREE | **42.3** | 47.1 | **37.2** | 2.1 | 0.9 | 3.2 | 23.1 | 27.5 | **18.9** | **16.5** | 20.9 | 13.0 |

Table 16: We use Real as the known domain ($\mathcal{T}_A$) and Infograph as the unknown domain ($\mathcal{T}_B$), and report clustering performance on $\mathcal{T}_A$, $\mathcal{T}_B$, as well as all other domains.

| Methods | Real | | | Infograph | | | others | | |
|---|---|---|---|---|---|---|---|---|---|
| | All | Old | New | All | Old | New | All | Old | New |
| RankStats+ | 34.2 | 62.4 | 19.6 | 12.5 | **21.9** | 6.3 | 18.5 | **32.1** | 6.4 |
| UNO+ | 42.8 | 69.4 | 29.0 | 10.9 | 15.2 | 8.0 | 18.2 | 28.0 | 9.6 |
| ORCA | 29.1 | 47.7 | 20.1 | 8.6 | 13.7 | 7.1 | 13.8 | 24.8 | 5.4 |
| GCD | 41.9 | 46.1 | 39.0 | 10.9 | 17.1 | 8.8 | 19.0 | 29.1 | 11.1 |
| SimGCD | 52.7 | 67.0 | 44.8 | 11.6 | 15.4 | 9.1 | 20.8 | 28.4 | 14.2 |
| SPTNet | 53.4 | 67.9 | 45.1 | 12.1 | 16.2 | 8.9 | 20.9 | 28.6 | 14.4 |
| RLCD | 53.9 | 68.3 | 45.8 | 12.5 | 16.7 | 9.1 | 21.5 | 29.4 | 14.6 |
| CDAD-Net | 53.5 | 65.6 | 47.2 | 13.6 | 17.2 | 9.8 | 22.0 | 29.1 | 16.2 |
| HiLo | 64.2 | 78.1 | 57.0 | 13.7 | 16.4 | 11.9 | 23.0 | 28.5 | 18.3 |
| FREE | **66.5** | **80.4** | **60.3** | **15.9** | 17.2 | **13.8** | **24.1** | 29.4 | **19.5** |

Table 17: Detailed clustering performance on other domains when using Real as $\mathcal{T}_A$ and Infograph as $\mathcal{T}_B$.

| Methods | Painting | | | Quickdraw | | | Sketch | | | Clipart | | |
|---|---|---|---|---|---|---|---|---|---|---|---|---|
| | All | Old | New | All | Old | New | All | Old | New | All | Old | New |
| RankStats+ | 29.6 | **49.2** | 10.0 | 2.5 | 1.6 | 3.4 | 17.4 | **32.2** | 6.5 | 24.4 | **45.5** | 5.8 |
| UNO+ | 30.8 | 44.8 | 16.8 | 2.7 | 2.3 | 3.1 | 17.0 | 27.0 | 9.7 | 22.3 | 37.8 | 8.7 |
| ORCA | 20.0 | 40.2 | 8.1 | 1.6 | 1.8 | 1.2 | 13.2 | 21.1 | 8.0 | 20.5 | 36.0 | 4.1 |
| GCD | 30.8 | 45.1 | 18.4 | **3.6** | **4.7** | 2.5 | 18.8 | 26.4 | 11.2 | 22.9 | 40.0 | 12.3 |
| SimGCD | 35.9 | 45.6 | 26.3 | 2.1 | 1.7 | 2.5 | 20.8 | 29.3 | 14.5 | 24.5 | 36.9 | 13.6 |
| SPTNet | 36.3 | 46.1 | 26.4 | 2.3 | 1.8 | 2.7 | 21.2 | 29.7 | 14.8 | 24.9 | 37.2 | 14.1 |
| RLCD | 37.2 | 46.8 | 26.9 | 2.5 | 1.9 | 2.9 | 22.3 | 30.5 | 15.3 | 25.1 | 37.3 | 14.7 |
| CDAD-Net | 39.3 | 46.5 | 29.6 | 2.5 | 1.7 | **3.4** | 21.8 | 30.5 | 16.1 | 25.6 | 36.5 | 16.4 |
| HiLo | 40.1 | 46.1 | 35.8 | 2.0 | 2.2 | 1.5 | 22.6 | 29.4 | 17.6 | 26.6 | 36.3 | 18.1 |
| FREE | **44.5** | 47.2 | **39.7** | 2.8 | 2.9 | 2.3 | **24.1** | 30.2 | **19.3** | **28.7** | 37.1 | **20.2** |

## B.5 Performance of different backbones

We further investigated the impact of using different backbone architectures on model performance. In particular, we adopted the pre-trained CLIP model [27] as our backbone and re-ran all the experiments accordingly. As shown in Table 18, the more advanced backbone consistently led to noticeable improvements across various evaluation metrics. Importantly, even with this stronger backbone, our proposed FREE framework continued to outperform other competing methods by a significant margin. This not only demonstrates the robustness of our approach but also provides additional evidence of its effectiveness and generalizability when combined with cutting-edge feature extractors.

Table 18: Clustering performance using different backbone architectures on DomainNet. We use Real as the known domain ($\mathcal{T}_A$) and Painting as the unknown domain ($\mathcal{T}_B$).

| Methods | Backbone | Real | | | Painting | | |
|---|---|---|---|---|---|---|---|
| | | All | Old | New | All | Old | New |
| HiLo | DINO | 64.4 | 77.6 | 57.5 | 42.1 | 42.9 | 41.3 |
| FREE | DINO | **67.7** | **78.1** | **61.2** | **45.6** | **46.1** | **44.8** |
| HiLo | CLIP | 74.5 | 78.1 | 64.2 | 47.1 | 49.5 | 45.4 |
| FREE | CLIP | **78.2** | **78.3** | **69.5** | **51.0** | **54.2** | **49.3** |

Table 19: Clustering performance using the estimated number of categories.

| Methods | $|C^u|$ | Original | | | Corrupted | | |
|---|---|---|---|---|---|---|---|
| | | All | Old | New | All | Old | New |
| HiLo | GT. (200) | 56.8 | 54.0 | 60.3 | 52.0 | 53.6 | 50.5 |
| FREE | GT. (200) | **60.4** | **58.5** | **63.2** | **55.7** | **57.1** | **53.7** |
| HiLo | Est. (257) | 55.9 | 52.9 | 59.2 | 51.2 | 52.8 | 49.5 |
| FREE | Est. (257) | **59.3** | **56.1** | **62.0** | **54.4** | **56.1** | **52.3** |

## B.6 Unknown category number

In previous experiments, we assumed that the number of categories was known in advance. Here, we extend our study to the more realistic scenario where the number of categories is unknown beforehand. To address this, we first employ the category estimation algorithm proposed in [33, 35] to perform an offline estimation of the number of categories. Based on these estimates, we rerun all experiments accordingly. As shown in Table 19, on CUB dataset, even when using the estimated number of categories, our proposed FREE framework consistently outperforms the strongest baseline method, HiLo. This result further validates the robustness and adaptability of our approach in practical settings where prior knowledge of the category count is unavailable.

## B.7 Parameter sensitivity analysis

Our proposed FREE framework involves several key hyperparameters. For certain parameters such as $\beta$ and $\epsilon$, we follow the settings adopted in prior work [38, 35]. For the temperature coefficients used in representation learning and clustering objectives, we follow the settings in [38, 35]. Specifically, the temperature $\tau$ is set to 0.07 for the self-supervised contrastive loss and 0.1 for the supervised contrastive loss. For the clustering loss, the temperature $\tau_s$ is set to 0.1, and $\tau_t$ is initialized at 0.07 and gradually warmed up to 0.04 during the first 30 epochs using a cosine scheduling strategy. Following [42], we define a square window with side length $2L * min(H, W)$ to select low-frequency components in the frequency domain. We conduct a sensitivity analysis on $L$. We use the Real domain as the known domain and Painting as the unknown domain. As shown in Table 20, the best performance is achieved when $L$ is set to 0.04. Moreover, we observe that the model maintains stable performance even with slight variations in this parameter, indicating that the proposed method is robust and relatively insensitive to the choice of $L$.

Next, we investigate the effect of the memory bank size $M$, which stores style information for contrastive and clustering objectives. As shown in Table 21, increasing the memory bank size generally improves model performance, as it allows the model to access a more diverse and representative set of style features. However, excessively large memory banks may incur additional computational and memory overhead. Based on empirical results, we find that a size of 1024 strikes a good balance between performance and efficiency.

Finally, we further investigate the impact of the number of nearest neighbors $K$ used in the domain separation strategy. As shown in Table 22, the performance of our model remains stable across a reasonable range of $K$ values.

Table 20: Parameter analysis of $L$

| $L$ | Real | | | Painting | | |
| --- | --- | --- | --- | --- | --- | --- |
| | All | Old | New | All | Old | New |
| 0.01 | 67.4 | 77.8 | 61.1 | 45.3 | 46.0 | 44.3 |
| 0.04 | **67.7** | **78.1** | **61.2** | **45.6** | **46.1** | **44.8** |
| 0.06 | 67.6 | 78.0 | 61.0 | 45.5 | 45.9 | 44.7 |
| 0.10 | 67.5 | 77.9 | 61.2 | 45.4 | 46.0 | 44.5 |

Table 21: Parameter analysis of memory bank size $M$

| $M$ | Real | | | Painting | | |
| --- | --- | --- | --- | --- | --- | --- |
| | All | Old | New | All | Old | New |
| 512 | 67.5 | 78.0 | 61.1 | 45.5 | 46.1 | 44.6 |
| 1024 | 67.7 | **78.1** | 61.2 | 45.6 | 46.1 | 44.8 |
| 2048 | **67.8** | 77.9 | **61.5** | **45.8** | **46.2** | **44.9** |

Table 22: Parameter analysis of nearest neighbors $K$

| $K$ | Real | | | Painting | | |
| --- | --- | --- | --- | --- | --- | --- |
| | All | Old | New | All | Old | New |
| 1 | 67.5 | 78.0 | 61.0 | 45.5 | 46.0 | 44.7 |
| 3 | **67.7** | **78.1** | **61.2** | **45.6** | **46.1** | **44.8** |
| 5 | 67.4 | 77.8 | 61.1 | 45.4 | 46.0 | 44.6 |

## B.8 Visualization of the attention map

To gain a deeper understanding of the advantages of our proposed FREE framework, we visualize the attention patterns within the final block of the ViT backbone. Specifically, we extract attention maps corresponding to the [CLS] token from all attention heads and highlight the top 10% most attended patches in red across the 12 attention heads. As shown in Fig. 5, our method consistently attends to semantically meaningful foreground regions, regardless of whether the samples come from known or unknown domains, and whether they belong to seen or unseen categories. This focused attention demonstrates that our method effectively suppresses distractions from background style variations, which are often domain-specific and irrelevant to semantic understanding. These observations further validate the effectiveness of the FREE framework in learning robust and domain-invariant representations.

## B.9 Incorporating various UDA techniques for DS_GCD

To explore whether recent advances in unsupervised domain adaptation (UDA) can benefit DS_GCD, we integrate several state-of-the-art UDA techniques—such as Mixup [40], FACT [39], and MixStyle [45]—into the SimGCD framework. As shown in Table 23, these additions result in only marginal performance improvements, indicating that directly applying existing UDA methods is insufficient for

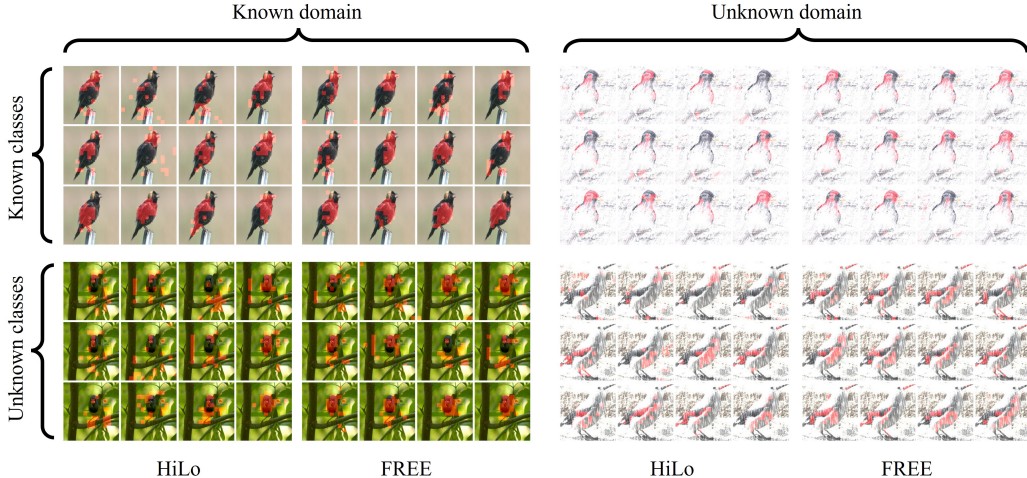

Figure 5: Attention visualization from the final block of the ViT backbone on the CUB-C dataset. The top 10% most attended patches from different heads are highlighted in red. Our method shows stronger focus on foreground regions across both known and unknown domains, demonstrating improved robustness to background style variations.

Table 23: Clustering performance of other UDA methods.

| Method | Real | | | Painting | | |
|--------|------|-----|-----|----------|-----|-----|
| | All | Old | New | All | Old | New |
| SimGCD | 61.3 | 77.8 | 52.9 | 34.5 | 35.6 | 33.5 |
| Mixup | 62.7 | 76.5 | 54.3 | 34.9 | 37.2 | 32.5 |
| Mixstyle | 62.3 | 76.8 | 54.0 | 35.0 | 36.1 | 34.0 |
| FACT | 62.9 | 76.9 | 55.1 | 36.1 | 37.5 | 34.9 |
| FREE | **67.7** | **78.1** | **61.2** | **45.6** | **46.1** | **44.8** |

Table 24: Clustering performance of other separation strategy.

| Method | Real | | | Painting | | |
|--------|------|-----|-----|----------|-----|-----|
| | All | Old | New | All | Old | New |
| Semi-supervised K-means | 65.8 | 77.7 | 58.1 | 42.8 | 45.7 | 42.1 |
| FREE | **67.7** | **78.1** | **61.2** | **45.6** | **46.1** | **44.8** |

solving DS_GCD. This further underscores the necessity and effectiveness of our proposed approach.

## B.10    Incorporating other separation strategy

We further compare our frequency-based domain separation strategy with alternative separation methods. Specifically, we adopt the semi-supervised K-means algorithm from [35] to divide the data into known and unknown domains, and then rerun the experiments. As shown in Table 24, our frequency-guided domain separation strategy achieves better performance across different domains, demonstrating its superiority.

## B.11    Stability analysis

To further assess the robustness and reproducibility of our framework, we evaluate the stability of model performance under different random initialization seeds. Specifically, each experiment is independently repeated three times with distinct random seeds, and the mean and standard deviation

of clustering accuracy are reported in Table 25 and Table 26. The results demonstrate that our method maintains highly stable performance across runs. This indicates that the proposed optimization procedure and domain-aware components do not introduce sensitivity to stochastic factors such as weight initialization or data sampling, confirming the consistency and reliability of our approach.

Table 25: Clustering performance of different methods on the SSB-C benchmark. Results are reported as mean $\pm$ std.

| Methods | CUB-C | | | | | | Scars-C | | | | | | FGVC-C | | | | | |
|---|---|---|---|---|---|---|---|---|---|---|---|---|---|---|---|---|---|---|
| | Original | | | Corrupted | | | Original | | | Corrupted | | | Original | | | Corrupted | | |
| | All | Old | New | All | Old | New | All | Old | New | All | Old | New | All | Old | New | All | Old | New |
| RankStats+ | 19.3±2.3 | 22.0±2.7 | 15.4±1.8 | 13.6±2.4 | 23.9±2.1 | 4.5±0.6 | 14.8±2.3 | 20.8±2.2 | 7.8±2.0 | 11.5±2.6 | 22.6±2.3 | 1.0±0.2 | 14.4±2.3 | 16.4±2.2 | 14.5±1.9 | 8.3±1.1 | 15.6±2.4 | 5.0±0.7 |
| UNO+ | 25.9±2.5 | 40.1±2.3 | 21.3±2.0 | 21.5±2.3 | 33.4±2.4 | 8.6±1.2 | 22.0±2.2 | 41.8±2.1 | 7.0±1.8 | 16.9±2.5 | 29.8±2.0 | 4.5±0.6 | 22.0±2.4 | 33.4±2.1 | 15.8±2.3 | 16.5±2.6 | 25.2±2.0 | 8.8±1.3 |
| ORCA | 18.2±1.6 | 22.8±2.4 | 14.5±1.9 | 21.5±2.0 | 23.1±2.3 | 18.9±2.2 | 19.1±2.2 | 28.7±2.0 | 11.2±1.6 | 15.0±2.4 | 22.4±2.1 | 8.3±1.0 | 17.6±1.9 | 19.3±2.2 | 16.1±2.1 | 13.9±1.4 | 17.3±1.8 | 10.1±2.2 |
| GCD | 26.6±1.1 | 27.5±1.7 | 25.7±1.3 | 25.1±1.0 | 28.7±1.4 | 22.0±1.2 | 22.1±1.6 | 35.2±1.3 | 20.5±1.1 | 21.6±1.5 | 29.2±1.1 | 10.5±1.4 | 25.2±1.5 | 28.7±1.2 | 23.0±1.4 | 21.0±1.3 | 23.1±1.0 | 17.3±1.6 |
| SimGCD | 31.9±2.3 | 33.9±1.9 | 29.0±2.0 | 28.8±2.4 | 31.6±2.1 | 25.0±2.0 | 26.7±2.2 | 39.6±2.1 | 25.6±1.9 | 22.1±2.5 | 30.5±2.1 | 14.1±2.4 | 26.1±2.3 | 28.9±2.0 | 25.1±1.9 | 22.3±2.1 | 23.2±2.3 | 21.4±2.0 |
| SPTNet | 33.0±1.7 | 34.5±1.1 | 31.2±1.9 | 30.1±2.0 | 33.1±1.4 | 26.1±2.2 | 28.0±1.5 | 40.2±2.0 | 27.9±1.6 | 24.2±2.3 | 32.1±1.8 | 16.3±1.3 | 28.7±2.0 | 30.2±1.7 | 27.9±2.2 | 24.8±1.5 | 25.7±1.1 | 23.9±2.4 |
| RLCD | 35.9±1.9 | 35.1±1.2 | 33.2±2.1 | 32.3±1.4 | 34.8±2.0 | 28.5±1.1 | 29.8±1.8 | 41.2±2.2 | 30.4±1.5 | 25.3±1.0 | 33.4±2.1 | 18.1±1.6 | 27.9±1.3 | 30.1±2.3 | 26.8±1.4 | 24.4±2.2 | 26.8±1.2 | 22.7±2.0 |
| CDAD-Net | 40.4±1.8 | 38.9±1.3 | 39.3±2.2 | 37.7±1.9 | 39.1±1.5 | 34.2±2.3 | 32.1±1.4 | 42.9±2.0 | 32.2±1.6 | 28.8±2.1 | 35.6±1.7 | 21.4±1.9 | 33.8±2.0 | 35.5±1.1 | 31.2±2.2 | 27.8±1.6 | 29.6±2.4 | 25.6±1.5 |
| HiLo | 56.8±1.6 | 54.0±1.3 | 60.3±1.5 | 52.0±1.6 | 53.6±1.5 | 50.5±1.6 | 39.5±1.2 | 44.8±1.7 | 37.0±1.3 | 35.6±1.8 | 42.9±1.2 | 28.4±1.5 | 44.2±1.7 | 50.6±1.6 | 47.4±1.3 | 31.2±1.4 | 29.0±1.5 | 33.4±1.6 |
| FREE | 60.4±1.9 | 58.5±2.1 | 63.2±2.0 | 55.7±2.2 | 57.1±1.8 | 53.7±1.4 | 43.6±1.9 | 48.1±2.1 | 40.8±1.7 | 38.9±1.5 | 46.1±1.8 | 32.6±1.2 | 48.5±2.0 | 54.9±1.9 | 51.2±1.6 | 35.0±1.3 | 32.4±1.7 | 38.9±1.5 |

Table 26: Clustering performance of different methods on the DomainNet benchmark. Results are reported as mean $\pm$ std.

| Methods | Real+Painting | | | | | | Real+Sketch | | | | | | Real+Quickdraw | | | | | | Real+Clipart | | | | | | Real+Infograph | | | | | |
|---|---|---|---|---|---|---|---|---|---|---|---|---|---|---|---|---|---|---|---|---|---|---|---|---|---|---|---|---|---|---|
| | Real | | | Painting | | | Real | | | Sketch | | | Real | | | Quickdraw | | | Real | | | Clipart | | | Real | | | Infograph | | |
| | All | Old | New | All | Old | New | All | Old | New | All | Old | New | All | Old | New | All | Old | New | All | Old | New | All | Old | New | All | Old | New | All | Old | New |
| RankStats+ | 34.1±2.1 | 62.0±2.8 | 19.7±2.4 | 29.7±2.2 | 49.7±2.3 | 9.6±0.9 | 34.2±2.5 | 62.0±2.9 | 19.8±2.3 | 17.1±2.6 | 31.1±2.8 | 6.8±1.1 | 34.1±2.7 | 62.5±2.5 | 19.5±2.0 | 4.1±1.2 | 4.4±0.7 | 3.9±0.8 | 34.0±2.2 | 62.4±2.8 | 19.4±2.1 | 24.1±2.4 | 45.1±2.2 | 6.2±2.0 | 34.2±2.3 | 60.4±2.5 | 29.0±2.6 | 12.5±1.9 | 21.9±2.5 | 6.3±1.0 |
| UNO+ | 44.2±2.4 | 72.2±2.8 | 29.7±2.0 | 31.1±2.3 | 45.1±2.5 | 17.2±1.1 | 43.7±2.2 | 72.5±2.8 | 28.9±2.5 | 12.5±1.9 | 17.0±2.4 | 9.2±1.3 | 31.1±2.5 | 60.0±2.7 | 16.1±1.2 | 6.3±1.4 | 5.8±0.6 | 6.8±0.6 | 44.5±2.1 | 66.1±2.9 | 33.3±2.3 | 21.9±2.4 | 35.6±2.7 | 10.1±2.0 | 42.8±2.3 | 69.4±2.5 | 29.0±2.6 | 10.9±2.0 | 15.2±2.5 | 8.0±1.3 |
| ORCA | 31.9±1.5 | 49.8±1.7 | 23.5±1.9 | 28.7±2.3 | 38.5±2.2 | 7.1±0.9 | 32.5±1.1 | 50.0±1.3 | 23.9±2.1 | 11.4±1.8 | 14.5±1.3 | 7.2±1.0 | 19.2±2.0 | 39.1±2.0 | 15.3±2.4 | 3.4±1.1 | 3.5±0.6 | 3.2±0.9 | 32.0±0.9 | 49.7±1.5 | 23.9±1.3 | 19.1±1.2 | 31.8±1.0 | 4.3±0.5 | 29.1±2.2 | 47.7±1.8 | 20.1±2.4 | 8.6±1.3 | 13.7±1.9 | 7.1±1.3 |
| GCD | 47.3±0.6 | 53.6±1.5 | 44.1±1.2 | 32.9±1.4 | 41.8±1.7 | 23.0±0.8 | 48.0±1.3 | 53.8±1.4 | 45.3±1.2 | 16.6±0.7 | 22.4±1.0 | 11.1±1.0 | 37.6±1.4 | 41.0±1.6 | 35.2±1.0 | 5.7±0.8 | 4.2±0.9 | 6.9±1.0 | 47.7±1.6 | 53.8±1.3 | 44.3±1.1 | 22.4±1.0 | 34.4±1.0 | 16.0±1.8 | 41.9±1.1 | 46.1±1.2 | 39.0±1.8 | 10.9±1.4 | 17.1±1.3 | 8.8±1.1 |
| SimGCD | 61.3±1.1 | 77.8±1.4 | 52.9±2.1 | 34.5±1.9 | 35.6±2.1 | 33.5±1.1 | 62.4±1.7 | 77.6±2.4 | 54.6±1.5 | 16.4±2.0 | 20.2±1.0 | 13.6±2.5 | 47.4±1.8 | 37.4±1.1 | 6.6±0.7 | 5.8±0.7 | 7.5±0.9 | 6.1±2.5 | 64.6±1.6 | 77.2±2.4 | 53.6±1.6 | 23.9±1.0 | 31.5±2.2 | 17.3±1.6 | 52.7±2.0 | 67.0±1.7 | 44.8±1.3 | 11.6±2.2 | 15.4±2.5 | 9.1±1.2 |
| SPTNet | 61.6±1.5 | 76.9±2.3 | 54.7±2.0 | 35.2±1.8 | 35.9±1.2 | 35.1±1.2 | 63.3±1.1 | 77.8±2.2 | 55.3±1.8 | 16.7±2.0 | 26.0±1.0 | 11.3±2.4 | 47.1±2.2 | 65.6±1.3 | 35.4±1.3 | 6.9±1.3 | 5.7±0.4 | 7.7±1.3 | 62.5±1.6 | 76.5±1.4 | 55.4±1.9 | 24.7±1.2 | 30.9±1.4 | 18.8±1.5 | 54.5±1.6 | 67.9±2.1 | 46.2±1.3 | 11.9±1.7 | 19.4±1.8 | 7.9±1.1 |
| RLCD | 62.1±1.9 | 78.3±1.2 | 53.8±1.1 | 36.9±2.3 | 35.7±2.4 | 36.2±2.1 | 62.8±1.4 | 77.4±1.1 | 55.7±2.0 | 17.0±1.6 | 20.4±1.2 | 15.2±1.7 | 49.1±1.0 | 67.8±2.3 | 38.0±1.4 | 7.0±0.9 | 5.8±0.6 | 7.8±1.2 | 62.3±1.8 | 77.1±1.3 | 54.7±2.4 | 24.5±2.1 | 38.0±2.3 | 13.9±2.3 | 57.2±1.8 | 68.3±2.3 | 48.1±1.1 | 12.0±1.3 | 15.9±1.1 | 9.8±0.9 |
| CDAD-Net | 63.6±1.5 | 77.8±1.4 | 56.3±2.2 | 38.4±1.5 | 38.4±1.4 | 37.5±1.8 | 61.9±1.2 | 76.3±2.1 | 52.1±1.1 | 17.3±2.4 | 20.9±2.1 | 15.9±1.3 | 48.5±1.0 | 66.5±2.1 | 36.7±2.0 | 6.4±1.0 | 5.6±0.7 | 7.3±1.1 | 61.3±1.5 | 77.0±1.3 | 53.1±1.8 | 25.2±1.6 | 31.9±1.4 | 19.0±2.2 | 56.5±2.0 | 68.0±1.4 | 47.1±1.3 | 11.8±1.2 | 15.6±1.7 | 9.4±0.8 |
| HiLo | 64.4±1.5 | 77.6±1.5 | 57.5±1.6 | 42.1±1.6 | 42.9±1.7 | 41.3±1.4 | 63.3±1.7 | 77.9±1.8 | 55.9±1.5 | 19.4±1.1 | 22.4±1.7 | 17.1±1.4 | 58.6±1.6 | 76.4±1.1 | 52.5±1.4 | 7.4±1.1 | 6.9±0.5 | 8.0±1.6 | 63.8±1.0 | 77.6±1.0 | 56.6±1.8 | 27.7±1.2 | 34.6±1.3 | 21.7±1.6 | 64.2±1.8 | 78.1±1.0 | 57.0±1.0 | 13.7±1.3 | 16.4±1.3 | 11.9±1.1 |
| FREE | 67.7±1.6 | 78.1±1.5 | 61.2±1.3 | 45.6±1.4 | 46.1±1.6 | 44.8±1.2 | 67.8±1.8 | 78.2±1.5 | 61.6±1.4 | 22.5±1.1 | 25.8±1.3 | 20.9±1.1 | 61.4±1.7 | 78.1±1.6 | 55.1±1.3 | 8.9±1.0 | 7.8±0.4 | 9.0±1.2 | 66.4±1.8 | 78.1±1.6 | 60.1±1.3 | 29.3±1.5 | 37.2±1.6 | 26.3±1.0 | 68.1±1.9 | 78.9±1.5 | 60.2±1.4 | 16.1±1.1 | 18.6±1.3 | 13.4±1.0 |

## C  Potential societal impact

This work focuses on Generalized Category Discovery (GCD) under distribution shift, a critical yet underexplored scenario where models must identify novel categories in unlabeled data while encountering significant domain changes. From a societal perspective, this line of research advances our ability to build adaptive AI systems that can reliably operate in evolving or previously unseen environments, which is essential in many real-world applications such as environmental monitoring, autonomous systems, medical diagnostics, and security surveillance.

By encouraging models to continuously discover and adapt to new concepts under domain shifts, our approach enhances the robustness and longevity of deployed machine learning systems, especially in non-stationary environments. For example, in healthcare, models must adapt to data from different hospitals or demographics without requiring exhaustive annotations. In safety-critical domains, failure to adapt to new distributions could lead to severe consequences, such as misdiagnosis or false alerts. However, with the ability to autonomously learn and categorize new concepts comes the risk of amplifying spurious correlations present in source domains. If the source domain data carries latent biases, these biases may propagate during the discovery process, particularly under domain shift, where spurious features may become more dominant. Thus, future work should explore bias detection and mitigation strategies tailored to open-world and shifting-domain settings. In summary, while our work enables more generalizable and context-aware AI, careful attention must be paid to ethical deployment, bias transfer, and accountability mechanisms, especially in high-stakes applications where model decisions evolve beyond initial human supervision.

