# OpenReview forum: "Generalized Category Discovery under Domain Shift: A Frequency Domain Perspective"
_NeurIPS.cc/2025/Conference — NeurIPS 2025 poster_

### Official Review · Reviewer_xWh6 · 2025-06-28

**Clarity:** 2
**Significance:** 3
**Originality:** 2
**Rating:** 4
**Confidence:** 4

**Summary:**

This paper focuses on generalized category discovery under domain shift, which is more practical compared to traditional UDA and GCD. To handle this scenario, this paper proposes a divide and conquer strategy based on the frequency domain, namely Free, which includes five objective functions.

**Questions:**

>1. Eq.(11) seems a probability, not an objective function, how to utilize it?
>2. What is the setting in Fig. 2?

**Ethical Concerns:**

["NO or VERY MINOR ethics concerns only"]

**Final Justification:**

The authors have addressed my concerns regarding the lack of motivation and missing related work, which were my main reasons for giving a borderline reject. I therefore raise my score to borderline accept.

**Limitations:**

The limitations can be found in the weaknesses and questions.

**Quality:**

2

**Strengths And Weaknesses:**

## Strengths

This paper tackles a more realistic scenario, and the reported results show noticeable improvements on the chosen benchmarks.


## Weaknesses


>1. Limited novelty in motivation and methodology.
>> The proposed framework is a divide-and-conquer integration of UDA and GCD techniques. From the perspective of motivation, this paper only states that “GCD fails under domain shift” without offering deeper insight. From the perspective of methodology, frequency-domain manipulation has been widely used in UDA [1, 2], and the corresponding components in the proposed Free also handle domain shift merely. As a result, the method may appear more like an extension of UDA rather than a principled solution to the core challenges of GCD.
>2. **Missing key related work.**
>> CDAD-Net [3] tackles essentially the same setting but is neither cited nor compared. The manuscript must clarify conceptual differences and provide empirical comparisons.
>3. Complex method without hyper-parameter analysis.
>> The proposed pipeline contains five components, yet no parameter-sensitivity study is provided. Are all hyper-parameters fixed to the values listed in the paper?
>4. Writing issues.
>>* *Unknown domains* is confusing, since these unlabeled domains are visible during training. Target domains would be clearer.
>>* The methodological description in the abstract and introduction is vague, where the rationale for each component is not evident.

[1] Kumar, Vikash, et al. "Improving domain adaptation through class-aware frequency transformation." International Journal of Computer Vision 131.11 (2023): 2888-2907.
[2] Kumar, Vikash, et al. "Caft: Class aware frequency transform for reducing domain gap." Proceedings of the IEEE/CVF International Conference on Computer Vision. 2021.
[3] Rongali, Sai Bhargav, et al. "CDAD-Net: Bridging Domain Gaps in Generalized Category Discovery." Proceedings of the IEEE/CVF Conference on Computer Vision and Pattern Recognition. 2024.

---

> ### Author Rebuttal · Authors · 2025-07-30
>
> ### **Q1:Limited novelty in motivation and methodology**
> Thank you for your thoughtful feedback. We respectfully argue that, while frequency-based techniques have been explored in the UDA community, their direct application to GCD is non-trivial and potentially harmful. GCD presents unique challenges: unlike UDA, it must discover novel categories in unlabeled target domains without access to any ground truth. This reliance on pseudo-labels makes the task particularly sensitive to semantic noise. As shown in Table 1, naïvely applying frequency perturbations without distinguishing between known and unknown domains leads to negative transfer, especially in the novel class space.
> To address this, our FDS module explicitly distinguishes known and unknown samples before perturbation, avoiding cross-domain contamination and enabling stable representation learning. This is essential in GCD, where noisy pseudo-supervision is the only guidance available for novel categories.
> Furthermore, our method introduces three novel components specifically tailored for GCD: Class-aware Cross-Domain Frequency Perturbation: It leverages the more familiar source-domain data in a class-aware manner to guide cross-domain feature learning while preserving semantic structures. Intra-Domain Frequency Perturbation: This improve generalization within the unknown domain. Clustering Difficulty-Aware Sampling: To our knowledge, this is the first method to incorporate difficulty-awareness into GCD. It adaptively rebalances training based on estimated category-level clustering difficulty, addressing the issue of uneven class separability that is critical in discovering novel categories.
> While our approach draws inspiration from UDA literature, it is designed to principally address the underexplored GCD problem under domain shift, with components carefully constructed to handle the semantic noise, lack of supervision, and class discovery difficulties that define GCD. We hope this clarification better communicates the principled design and GCD-specific motivation behind our work.
>
> ### Table 1: Ablation study on different transformation strategies
> | Method     | Real All | Real Old | Real New | Painting All | Painting Old | Painting New |
> |-|-|-|-|-|-|-|
> | SimGCD     | 61.3  | 77.8 | 52.9 | 34.5| 35.6  | 33.5   |
> | Random     | 62.7 | 76.5  | 54.3 | 33.2 | 34.5  | 32.5 |
> | FREE | 67.7  | 78.1 | 61.2 | 45.6 | 46.1 | 44.8 |
>
> ### Table 2: Ablation study of different components.
> | FDS | IDFP | CDFP | CDAS | Real All | Real Old | Real New | Painting All | Painting Old | Painting New |
> |-|-|-|-|-|-|-|-|-|-|
> | ✗   | ✗    | ✗    | ✗    | 61.3     | 77.8     | 52.9     | 34.5  | 35.6  | 33.5  |
> | ✓   | ✓    | ✗    | ✗    | 62.0     | 77.8     | 53.3     | 36.6 | 37.8| 35.5|
> | ✓   | ✗    | ✓    | ✗    | 65.6     | 77.9     | 58.2     | 41.6 | 41.9 | 40.7 |
> | ✓   | ✗    | ✗    | ✓    | 63.1     | 77.9     | 54.8     | 37.6 | 38.0| 36.9  |
> | ✓   | ✓    | ✓    | ✗    | 66.1     | 78.0     | 59.5     | 42.8  | 43.1 | 42.0  |
> | ✓   | ✓    | ✓    | ✓    | 67.7     | 78.1     | 61.2     | 45.6  | 46.1 | 44.8 |
>
> ### **Q2:Missing key related work.**
> Thank you for the suggestion. While CDAD-Net tackles a related problem, it assumes that all unlabeled data come from unknown domains—a more restrictive setting. In contrast, our method addresses a more practical scenario where unlabeled data include both known and unknown domains. As shown in Tables 3 and 4, our approach consistently outperforms CDAD-Net across all metrics, highlighting its effectiveness in this more challenging setting. We will clarify the conceptual differences and emphasize the empirical results in the final version.
>
> ### Table 3: Performance on the SSB-C Dataset
> | Method     | CUB-O All | CUB-O Old | CUB-O New | CUB-C All | CUB-C Old | CUB-C New | Scars-O All | Scars-O Old | Scars-O New | Scars-C All | Scars-C Old | Scars-C New | FGVC-O All | FGVC-O Old | FGVC-O New | FGVC-C All | FGVC-C Old | FGVC-C New |
> |-|-|-|-|-|-|-|-|-|-|-|-|-|-|-|-|-|-|-|
> | CDAD-Net     | 40.4±1.8   | 38.9±1.3   | 39.3±2.2   | 37.7±1.9   | 39.1±1.5   | 34.2±2.3   | 32.1±1.4   | 42.9±2.0   | 32.2±1.6   | 28.8±2.1   | 35.6±1.7   | 21.4±1.9   | 33.8±2.0   | 35.5±1.1   | 31.2±2.2   | 27.8±1.6   | 29.6±2.4   | 25.6±1.5   |
> | FREE  | 60.4±1.1  | 58.5±1.4  | 63.2±1.3  | 55.7±1.6  | 57.1±1.7  | 53.7±1.5  | 43.6±1.1     | 48.1±1.5     | 40.8±1.2     | 38.9±1.7     | 46.1±1.6     | 32.6±1.5     | 48.5±1.3    | 54.9±1.4    | 51.2±1.1    | 35.0±1.5    | 32.4±1.3    | 38.9±1.2    |
>
> ### Table 4a: Real+Painting Performance on DomainNet
> | Method     | Real All       | Real Old       | Real New       | Painting All   | Painting Old   | Painting New   |
> |-|-|-|-|-|-|-|
> | CDAD-Net  | 63.6 ± 1.5   | 77.8 ± 1.4   | 56.3 ± 2.2   | 38.4 ± 1.5   | 38.4 ± 1.4   | 37.5 ± 1.8   |
> | FREE | 67.7±1.4       | 78.1±1.7       | 61.2±1.2       | 45.6±1.1       | 46.1±1.3       | 44.8±1.6       |
>
> ### Table 4b: Real+Sketch Performance on DomainNet
> | Method     | Real All       | Real Old       | Real New       | Sketch All   | Sketch Old   | Sketch New   |
> |-|-|-|-|-|-|-|
> | CDAD-Net  | 61.9 ± 1.2   | 76.3 ± 2.1   | 52.1 ± 1.1   | 17.3 ± 1.5   | 20.9 ± 1.1   | 15.9 ± 1.3   |
> | FREE | 67.8±1.5        | 78.2±1.4        | 61.6±1.3        | 22.5±1.4        | 25.8±1.3        | 20.9±1.1        |
>
> ### Table 4c: Real+Quickdraw Performance on DomainNet
> | Method     | Real All       | Real Old       | Real New       |Quickdraw All   | Quickdraw Old   | Quickdraw New   |
> |-|-|-|-|-|-|-|
> | CDAD-Net | 48.5 ± 1.0   | 66.5 ± 2.1   | 36.7 ± 2.0   | 6.4 ± 1.0    | 5.6 ± 1.1    | 7.3 ± 1.1    |
> | FREE | 61.4±0.9        | 78.1±1.6        | 55.1±1.8        | 8.9±0.7         | 7.8±0.5        | 9.0±0.9         |
>
> ### Table 4d: Real+Clipart Performance on DomainNet
> | Method     | Real All       | Real Old       | Real New       | Clipart All   | Clipart Old   | Clipart New   |
> |-|-|-|-|-|-|-|
> | CDAD-Net | 61.3 ± 1.5   | 77.0 ± 1.3   | 53.1 ± 1.8   | 25.2 ± 1.6   | 31.9 ± 1.4   | 19.0 ± 2.2   |
> | FREE | 66.4±1.0   | 78.1±0.9  | 60.1±0.9  | 29.3±1.5 | 37.2±1.6 | 26.3±1.6 |
>
> ### Table 4e: Real+Infograph Performance on DomainNet
> | Method | Real All  | Real Old | Real New | Infograph All   | Infograph Old   |Infograph New   |
> |-|-|-|-|-|-|-|
> | CDAD-Net  | 56.5 ± 2.0   | 68.0 ± 1.4   | 47.1 ± 1.3   | 11.8 ± 1.2   | 15.6 ± 1.7   | 9.4 ± 1.3    |
> | FREE  | 68.1±1.4 | 78.9±1.3 | 60.2±1.6 | 16.1±1.6 | 18.6±1.2   | 13.4±0.9        |
>
> ### **Q3:hyper-parameter analysis**
> Thank you for your comment. We would like to clarify that the sensitivity analysis of the hyperparameters designed in our framework is included in the appendix. As shown in the results, our method demonstrates stable performance across a wide range of settings, indicating its robustness to hyperparameter choices.
> In addition, we now include a new ablation study on the loss balancing coefficient \\(\beta\\), and rewrite the overall loss function as: \\(L_{total} = \beta L_{kd} + (1 - \beta)L_{ud}+ \epsilon\\bigtriangleup\\). As shown in Table 5, setting $\beta$ to 0 or 1 leads to suboptimal performance. In contrast, our method consistently achieves stable results across different values of \\(\beta\\), further demonstrating the robustness of the proposed training strategy. We will ensure all parameter values used in experiments are clearly listed in the final version.
>
> ### Table 5: Sensitivity analysis of the loss weight $\beta$
> | \\(\beta\\) | Real All       | Real Old       | Real New       | Painting All   | Painting Old   | Painting New   |
> |-|-|-|-|-|-|-|
> | 0.0  |64.0 | 77.9 | 55.7 |39.3 | 40.2| 38.9|
> | 0.3 | 67.4 | 77.8 | 60.9 |45.3 | 45.8| 44.3|
> | 0.5| 67.7 | 78.1 | 61.2 |45.6 | 46.1| 44.8|
> | 0.7| 67.5 | 77.7 | 61.1 | 45.4 | 46.0| 44.4|
> | 1.0 | 66.5 | 77.9 | 59.8|42.8 | 44.0 | 42.1 |
>
> ### **Q4:Writing Clarity Concern**
> Thank you for pointing this out. We agree that the term unknown domain may be confusing since the unlabeled target domain is visible during training. In the final version, we will revise the terminology to use target domain consistently for clarity.
> We also acknowledge that the methodological descriptions in the abstract and introduction could be more precise. We will revise these sections to better articulate the intuition and motivation behind each component of our framework. These changes will improve the overall readability and clarity of the paper.
>
> ### **Q5:Eq.(11) seems a probability, not an objective function, how to utilize it?**
> We first sample categories from the categorical distribution \\(c \sim p_{\text{difficulty}}\\), where the probability \\(p_{\text{difficulty}}^c\\) reflects the clustering difficulty of class \\(c\\). We then retrieve real feature embeddings from the current pool of samples predicted as class \\(c\\) by the model. These features are used for classification or contrastive learning to refine the model's representation. We appreciate the feedback and will incorporate further details in the final version of the paper.
>
> ### **Q6:What is the setting in Fig. 2**
> Figure 2 presents the results of directly applying FDA to SimGCD, where we report the clustering accuracy across all categories. As shown, our method outperforms SimGCD+FDA both in terms of final clustering accuracy and convergence speed. In contrast, the random transformations introduced by SimGCD+FDA can sometimes hinder learning in the unknown domain, potentially leading to negative transfer effects. This observation further motivates the need for a more tailored perturbation strategy.

---

> > ### Author Response · Authors · 2025-08-05
> > **Official Comment by Authors**
> >
> > Dear Reviewer xWh6,
> >
> > We sincerely appreciate the time and effort you have invested in reviewing our submission. Your insightful feedback has been invaluable to us, and we have diligently worked to address all the concerns you raised in our rebuttal. As the author-reviewer discussion phase is drawing to a close, we would like to confirm whether our responses have effectively addressed your concerns. We are more than happy to provide any further details or explanations. Thank you once again for your thoughtful review and consideration.
> >
> > Best regards,
> >
> > The Authors

---

> > ### Comment · Reviewer_xWh6 · 2025-08-05
> >
> > I appreciate the authors for providing extensive experimental results and a detailed rebuttal.
> >
> > I have read the authors' response as well as comments from other reviewers, and I will temporarily maintain my score. The main reasons are as follows:
> > 1. **Missing discussion of relevant prior work**
> > The authors clarified that the CDAD-Net setting is largely aligned with theirs, which suggests that the combination of domain shift and GCD has already been studied. However, this prior work is not cited or discussed in the main paper, limiting the novelty of the proposed setting, specifically regarding the domain shift aspect, rather than the source of unlabeled data.
> > 2. **Lack of clearly articulated motivation**
> > While I acknowledge the motivation provided in the rebuttal, it is not clearly presented in the main paper. The core motivation of the work could be summarized as the failure of frequency-domain manipulation under DS-GCD settings, yet the paper only attributes it to general negative transfer. This explanation appears too broad and lacks sufficient insight.  Furthermore, as noted above, the paper lacks a deeper comparison with other domain shift methods, such as CDAD-Net, which could have better justified its uniqueness. Overall, I believe the contribution lies more in a stacked integration of components rather than a fundamental analysis of the DS-GCD problem itself.

---

> ### Author Response · Authors · 2025-08-07
>
> We sincerely thank the reviewer for their constructive feedback and for acknowledging the added experiments. We would like to further clarify the novelty, motivation, and technical contributions of our work:
>
> ---
>
> ### On CDAD-Net and Domain Shift under GCD Settings
>
> While **CDAD-Net** adopts a similar problem setting combining domain shift and generalized category discovery (GCD), we would like to emphasize that this problem remains far from being solved. For instance, on the **CUB** dataset, CDAD-Net achieves only **40.4%** accuracy on all classes in the original domain, and merely **37.7%** in the corrupted domain. In contrast, our method outperforms CDAD-Net by over **20.0%** on the original domain and **18.0%** on the corrupted domain across all classes. We attribute this performance gap to the **instability of CDAD-Net's entropy-driven adversarial learning strategy**, which motivated us to seek a more stable and principled solution.
>
> ---
>
> ### Motivation and Failure of Frequency Perturbation in DS-GCD
>
> As shown in Figure 2 (Page 2, Lines 51–60), we clearly demonstrate that applying standard frequency-domain perturbation (as used in UDA methods) leads to negative transfer in the DS-GCD setting. This is because indiscriminate frequency manipulation introduces semantic noise, which is especially harmful to novel class discovery, where pseudo-labels are the only source of supervision. This makes the task highly sensitive to such noise.
>
> To address this, our **FDS module** explicitly distinguishes between **known and unknown domain samples** before perturbation, avoiding **cross-domain contamination** and enabling **stable representation learning**.
>
> ---
>
> ### Clarification on Paper Motivation and Component Design
>
> Each component of our method is designed with specific motivations tailored for the DS-GCD challenge:
>
> - **Section 4.2 (Lines 190–197)** motivates the need for *class-aware cross-domain frequency perturbation*, which leverages more stable source-domain data in a category-aware manner to guide feature adaptation while preserving semantic structures.
>
> - **Section 4.3 (Lines 211–217)** explains *intra-domain frequency perturbation*, which improves generalization within the target domain.
>
> - **Section 4.4 (Lines 230–237)** introduces *clustering difficulty-aware sampling*, which rebalances training by estimating *class-level clustering difficulty* — a factor not yet explored in prior GCD literature.
>
> Additionally, the contents of Table 1 and Table 2 in the rebuttal correspond to Table 3 and Table 4 in the main paper, respectively.
> We agree that the motivation could be further strengthened in the main text. In the final version, we will **revise and enrich** the discussion in both the **introduction** and **method** sections to highlight our **principled design** and the rationale behind each component.
>
> ---
>
> ### CDAD-Net Discussion and Citation
>
> Thank you for the suggestion — we will include a detailed discussion of CDAD-Net in the final version. For clarity:
>
> > “CDAD-Net achieves cross-domain known class alignment and novel class discovery by integrating entropy-driven adversarial learning, cross-domain prototype alignment, neighborhood contrastive learning, and conditional image inpainting to jointly optimize a shared representation space.”
>
> In contrast to CDAD-Net, **our approach is the first to introduce a frequency-domain solution tailored for the unique challenges posed by domain-shifted generalized category discovery (DS-GCD).**
>
> Although our method draws inspiration from UDA literature, it is fundamentally designed to tackle unaddressed challenges in GCD under domain shift, such as semantic noise, lack of supervision, and uneven cluster separability. We believe our contributions — including FDS, Class-aware Cross-Domain Frequency Perturbation, Intra-Domain Frequency Perturbation, and Clustering Difficulty-Aware Sampling — offer a novel and effective framework to address these unique challenges.
>
> ---
>
> We hope this clarifies our motivation and design principles, and we sincerely thank the reviewer for encouraging us to present them more clearly.

---

> > ### Comment · Reviewer_xWh6 · 2025-08-08
> >
> > Thank the authors for the detailed rebuttal. The analysis provided in the response, particularly the point "a factor not yet explored in prior GCD literature",  is valuable but appears too far back in the manuscript. These analyses noted in the rebuttal should be incorporated and highlighted earlier in the revised version. I will update my score.

---

> ### Author Response · Authors · 2025-08-08
>
> Thank you very much for your thoughtful and detailed review throughout this process. We deeply appreciate your recognition of our efforts in addressing the concerns you raised, as well as your acknowledgement that the changes and additions will greatly improve the paper. Your professional and constructive approach has been invaluable in helping us strengthen our work, and it is truly our honor to receive your guidance.
>
> As previously stated, our paper has already articulated the motivation and design objectives clearly in multiple sections, including both the **Introduction** and the **Method** parts. We also appreciate your suggestion regarding the placement of the analysis — especially the point on “a factor not yet explored in prior GCD literature.” We fully agree that this perspective is valuable and should appear earlier in the manuscript to strengthen our motivation. In the revised version, we will integrate these analyses into the **Introduction** and early parts of the **Method** section, ensuring they are highlighted and clearly connected to our contributions.
>
> **Overall, the failure of approaches such as CDAD-Net has prompted us to reflect more deeply on whether frequency-domain information could be leveraged to address the DS-GCD problem. Our method is specifically designed to explore this direction, providing a principled and effective solution that mitigates semantic noise, enhances cross-domain generalization, and facilitates robust novel category discovery under domain shift.**
>
> We are grateful for the time and expertise you invested in evaluating our research and for your valuable contributions to improving our paper.

---

### Official Review · Reviewer_ivLL · 2025-06-30

**Clarity:** 2
**Significance:** 3
**Originality:** 3
**Rating:** 4
**Confidence:** 4

**Summary:**

This paper presents a frequency-based approach to Generalized Category Discovery (GCD) that is designed to be robust under domain shift. The method consists of four main components. First, it separates known and unknown samples using a Gaussian Mixture Model (GMM). Next, it introduces Cross-Domain Frequency Perturbation (CDFP), which applies Fourier-based augmentations to samples of the same category across different domains, encouraging the model to generalize across domain boundaries. The third component, Intra-Domain Frequency Perturbation (IDFP), enhances robustness by applying similar augmentations within a single domain. Lastly, the method incorporates Difficulty-Aware Sampling, enabling the model to focus more on challenging samples during clustering.

**Questions:**

1- Regarding Table 6 in the appendix, GMM is set to two modes in the main paper, which aligns with separating known and unknown categories. However, in the context of the appendix, there appear to be more than just two relevant categories, and the distributions seem to differ significantly. Since there are no experiments directly demonstrating the impact of this GMM setup with different mode numbers, does this table suggest that using GMM in this way may not be effective?

2- Since the proposed difficulty-aware clustering and Fourier augmentation are not inherently limited to scenarios involving domain shift, what would be their impact when applied to a standard GCD baseline? It would be interesting to see if these components also benefit generalized category discovery in the absence of domain adaptation.

**Ethical Concerns:**

["NO or VERY MINOR ethics concerns only"]

**Final Justification:**

While I would ideally like to reproduce the results using the authors’ provided source code (in the rebuttal), the limited time window may make this difficult. Nevertheless, the authors have provided a detailed rebuttal, and I am in favor of accepting this paper.

**Limitations:**

While the authors acknowledge that their work is currently limited to image classification, it would also be valuable to discuss some method-specific limitations. For example, the use of FFT introduces additional computational overhead, which may impact scalability. Additionally, the effectiveness of the GMM component relies on its ability to accurately separate known and unknown categories, which may not always hold in practice. Addressing these potential limitations would provide a more balanced perspective on the approach.

**Paper Formatting Concerns:**

One formatting issue I noticed is in line 458 of the appendix. While this may not be explicitly covered in the formatting guidelines, I believe these styling inconsistencies should be addressed.

**Quality:**

3

**Strengths And Weaknesses:**

**Strengths:**

1- The method leverages Fourier-based frequency perturbations for both inter-domain (same class across domains) and intra-domain (within-domain) settings, which enhances the model’s ability to generalize under domain shift.

2- The introduction of Difficulty-Aware Sampling is a useful addition, helping the model focus on more challenging categories and potentially improving clustering performance across a range of scenarios.

3- The approach demonstrates effectiveness across multiple state-of-the-art baselines and includes extensive ablation studies (detailed in the appendix), supporting the robustness of the proposed contributions.


**Weaknesses:**

1- Although the paper claims five key contributions, several of them do not appear to be particularly major novel contributions:

FDS (Frequency-based Domain Separation): The use of a Gaussian Mixture Model (GMM) to separate known and unknown domains is introduced without clear justification or analysis of its necessity. It remains unclear how much this component contributes to the overall performance. While a semi-supervised K-means variant is briefly mentioned, it is not thoroughly explored or compared. Furthermore, it is not specified whether this clustering was performed in the frequency domain or on raw images, an important distinction, as K-means might perform better than GMM in image space.

IDFP (Intra-Domain Frequency Perturbation): This is essentially a basic augmentation strategy that has been explored in prior work. It is not clearly tied to the specific goals of Generalized Category Discovery, limiting its significance as a standalone contribution.

2- The related work section is lacking in important citations, both in the context of Generalized Category Discovery and frequency-based methods. For example, in line 112, the phrase "recent research has explored frequency-based domain adaptation strategies" is vague and does not cite any specific papers. Related work should explicitly reference prior studies to provide proper context, and prior methods such as ORCA are missing entirely from the related works section while being reported in experiments.

3- For a method composed of multiple components, the ablation studies in the main paper are limited, and the experiments in the appendix are introduced without sufficient explanation or discussion. This makes it difficult to understand the individual contribution of each module or justify design choices.

4- The writing could be improved for clarity. While the general structure is understandable, figures and tables are not always well explained, and several new terms and notations are introduced without clear definitions, which makes the paper more difficult to follow and interpret.

*Minor:*
Consider using an acronym (e.g., FREE) or consistently formatting the method name in bold or italics throughout the paper. Using “Free” in plain text makes it harder to distinguish from regular words.

The distinction between losses introduced by SimGCD and standard GCD losses is unclear. For clarity, it would help to explicitly label which components originate from prior GCD frameworks and which are specific to SimGCD.

The method mentions swapping low-frequency amplitudes, but it does not clearly define what constitutes "low frequency." Specifying a threshold or frequency range would provide a better understanding. Although there is some mention of low frequency in the appendix, it would be helpful to explicitly state in the main text how low-frequency components are identified and how the swapping is implemented, or to clearly direct readers to the relevant section in the appendix.

There are several notations introduced that are used only once, which adds unnecessary complexity. Simplifying these would improve readability.

Line 239: The notation $E(x_i)$ is used without definition; please clarify what this refers to.

Lines 261 and 264: It may be helpful to explicitly mention “HiLo” here so that readers understand the context without backtracking.

Figure 3: The figure would benefit from a clearer explanation, both in the caption and in the main text. Specifically, mapping each part of the diagram to its corresponding loss component would improve interpretability.

Line 333: "hindering" -> "hinders."

Line 447: The term "Theoretical Analysis" is misleading here; the content largely consists of background explanation on Fourier transforms. Consider renaming this section to Preliminaries for accuracy.

Line 458: The phrase “relative strength between different directional templates” is ambiguous. Additionally, please ensure that all formulas and technical statements are carefully reviewed for accuracy. The supplementary material, particularly in lines 458, 460, 463, and 465, sometimes lacks clarity and displays inconsistent style. If these sections were drafted with the help of AI tools, please carefully review them.


Line 472: The paper frequently omits proper citations. For example, DomainNet should be cited.

A pseudocode explaining the method in the appendix can be helpful to understand the steps well.

**Quality:**
While the motivation and core insights of the method are strong, the paper would benefit from further revision and refinement, especially in the appendix.

**Clarity:**
Some explanations lack clarity, and the use of notation is inconsistent, sometimes underexplained, other times overly detailed, which can make the paper challenging to follow.

**Significance:**
The method enhances domain adaptability through augmentation. Given that HiLo has already addressed domain shift within generalized category discovery, this work primarily positions itself at the intersection of domain adaptation and generalized category discovery, and more as a solution to an already established problem.

**Originality:**
Although Fourier amplitude swapping has been explored in domain generalization and adaptation, its application to generalized category discovery is novel. Additionally, the introduction of difficulty-aware sampling is an interesting contribution.

---

> ### Author Rebuttal · Authors · 2025-07-30
>
> ### **Q1:contribution of each component:**
> We thank the reviewer for the insightful comment. As shown in Table 1, directly applying frequency transformations without domain separation often leads to negative transfer, especially in the unseen domain where pseudo-labels are noisy. This is because the model relies entirely on pseudo-labels to supervise clustering in these unfamiliar domains, which may result in noisy guidance and degraded performance. This motivated us to first perform domain separation and leverage the more familiar source domain to learn domain-invariant feature representations. Building on this, we design two specialized frequency-domain perturbation strategies to improve domain generalization and reinforce semantic structure. As shown in Table 2, the class-aware cross-domain frequency perturbation enhances generalization by promoting domain-invariant representation learning. Additionally, the intra-domain frequency perturbation consistently boosts performance across all, old, and new classes in the target domain, reflecting improved intra-class compactness and inter-class separability. Finally, our CDAS module adaptively emphasizes harder categories based on clustering difficulty, leading to better representation learning and overall clustering quality.
>
> #### Table 1: Ablation study on different transformation strategies
> | Method     | Real All | Real Old | Real New | Painting All | Painting Old | Painting New |
> |-|-|-|-|-|-|-|
> | SimGCD     | 61.3 | 77.8 | 52.9 | 34.5 | 35.6 | 33.5   |
> | Random     | 62.7 | 76.5 | 54.3     | 33.2  | 34.5 | 32.5 |
> | FREE       | 67.7 | 78.1 | 61.2     | 45.6 | 46.1  | 44.8  |
>
> #### Table 2: Ablation study of different components
> | FDS | IDFP | CDFP | CDAS | Real All | Real Old | Real New | Painting All | Painting Old | Painting New |
> |-|-|-|-|-|-|-|-|-|-|
> | ✗   | ✗    | ✗    | ✗    | 61.3 | 77.8 | 52.9| 34.5| 35.6 | 33.5 |
> | ✓   | ✓    | ✗    | ✗    | 62.0 | 77.8  | 53.3| 36.6| 37.8 | 35.5  |
> | ✓   | ✗    | ✓    | ✗    | 65.6 | 77.9  | 58.2  | 41.6 | 41.9  | 40.7 |
> | ✓   | ✗    | ✗    | ✓    | 63.1 | 77.9  | 54.8  | 37.6| 38.0| 36.9  |
> | ✓   | ✓    | ✓    | ✗    | 66.1| 78.0 | 59.5  | 42.8  | 43.1  | 42.0   |
> | ✓   | ✓    | ✓    | ✓    | 67.7| 78.1  | 61.2 | 45.6  | 46.1  | 44.8 |
>
> Semi-supervised K-means was also conducted in the frequency domain. Additionally, we applied it to the original image space for comparison. The advantage of working in the frequency domain lies in the ability to isolate the amplitude component, which predominantly captures image style—something not feasible in the raw image space. As shown in Table 3, our method consistently outperforms Semi-supervised K-means in both domains. Frequency-based features yield stronger separation than raw image space representations under both clustering schemes.
>
> #### Table 3: Clustering performance of other separation strategies
> | Method     | Real All | Real Old | Real New | Painting All | Painting Old | Painting New |
> |-|-|-|-|-|-|-|
> | Semi-supervised K-means(image space) | 65.3 | 77.4 | 57.8 | 42.6 | 45.5 | 41.6 |
> | FREE | 67.7 | 78.1 | 61.2 | 45.6 | 46.1 | 44.8 |
>
> While IDFP alone has been explored in prior work as a general augmentation strategy, we observed that applying IDFP without the proposed FDS module leads to negative transfer and limited performance gains. Therefore, identifying a task-specific and well-integrated solution is crucial. Without the introduction of the FDS module, IDFP cannot effectively contribute to the generalized category discovery task.
>
> ### **Q2:lacking in important citations**
>
> We thank the reviewer for the valuable suggestion. We will add the following citation to clarify the statement in Line 112:
> > Yang, Yanchao, and Stefano Soatto. *FDA: Fourier Domain Adaptation for Semantic Segmentation*. CVPR 2020.
>
> This work serves as a key reference for frequency-based domain adaptation strategies. Additionally, we acknowledge the omission of prior methods such as ORCA in the related work section. We will include a more comprehensive discussion of these methods in the final version to ensure completeness.
>
> ### **Q3:ablation studies of each component**
> We appreciate the reviewer’s concern regarding the complexity of our framework and the need for clearer justification of each component. To address this, we have conducted comprehensive ablation studies on the DomainNet dataset, as summarized in Table 1 and Table 2. Starting from SimGCD as the baseline, we incrementally integrate each module and evaluate its individual impact. As shown in Table 1, applying frequency perturbation without domain separation often causes performance degradation, especially on the unknown domain. This is because performing perturbations indiscriminately in the frequency domain may lead to negative transfer. To mitigate this, we introduce domain separation, which leverages amplitude discrepancies across sample groups to isolate known and unknown domains. We then apply Cross-Domain Frequency Perturbation (CDFP) to guide domain-invariant representation learning using the more familiar source domain. Furthermore, we employ In-Domain Frequency Perturbation (IDFP) within the target domain to capture intra-domain variation while avoiding semantic collapse. As shown in Table 2, IDFP consistently improves clustering accuracy across all/known/unknown subsets. Finally, the Clustering Difficulty-Aware Sampling strategy further enhances performance by dynamically balancing training on easy and hard samples.
> These ablation results clearly demonstrate the necessity and complementary nature of each module. We will also revise the appendix to include more detailed explanations of the experimental setup and the design rationale for each component.
>
> ### **Q4:Writing clarity needs improvement**
>
> Thank you for the helpful suggestion. We will revise the manuscript to improve the clarity of figure captions and explanations. All terminology and symbols will be properly defined, and we will ensure consistency throughout the paper.
>
> ### **Q5:Ambiguity of the method name "Free"**
>
> We have changed Free to FREE and will apply consistent formatting throughout the final version.
>
> ### **Q6:Unclear difference between SimGCD and standard GCD losses**
>
> The contrastive loss is inherited from the original GCD framework, while the clustering loss is novel and specific to SimGCD. We will clarify this distinction in the revised version.
>
> ### **Q7:"Low frequency" not clearly defined**
>
> Following FDA, we define low-frequency components using a centered square of side length $2L*min(H,W)$ in the frequency domain. We will clarify this in the main text and refer readers to the corresponding analysis in the appendix.
>
> ### **Q8:Undeclared or rarely used symbols.For example, symbol $E$ is used without prior definition.**
> Thank you. The symbol $E$ denotes the feature extractor, which is introduced in the “Revisiting SimGCD” section. We will define it clearly when first used and simplify any unnecessarily complex notations.
>
> ### **Q9:Clarify references to "HiLo" in lines 261 and 264**
>
> Thanks. We will explicitly mention HiLo in those lines to improve clarity and minimize backtracking for readers.
>
> ### **Q10:Insufficient explanation of Figure 3**
>
> We appreciate the suggestion. We will revise both the caption and the main text to map each visual element to its corresponding loss component and enhance interpretability.
>
> ### **Q11:Grammatical error at line 333 ("hindering")**
>
> Thanks for pointing this out. We will correct the grammar in the final version.
>
> ### **Q12:Misleading section title "Theoretical Analysis"**
>
> We agree and will rename the section to Preliminaries for clarity and accuracy.
>
> ### **Q13:Ambiguity in line 458 and inconsistent supplementary content**
>
> Thank you. We will revise the supplementary material (especially lines 458, 460, 463, and 465) for clarity and consistency, and ensure the accuracy of all formulas and statements.
>
> ### **Q14:Missing citations (e.g., DomainNet at line 472)**
>
> Thank you. We will add all missing citations, including DomainNet, in the final version.
>
> ### **Q15:Suggestion to include pseudocode in the appendix**
>
> Thank you. We will include clear pseudocode in the appendix to improve method transparency and reproducibility.
>
> ### **Q16:GMM validity in multi-domain setting (Table 6). Does using a 2-mode GMM remain valid when multiple domains are present?**
> Even in multi-domain settings, a two-component GMM remains effective because all unknown domains are stylistically dissimilar to the known domain. As shown in Table 6, our method still performs well with multiple domains, validating the approach.
>
> ### **Q17:Applicability of components to standard GCD without domain shift**
>
> Thank you for the insightful question. When domain shift is absent, our modules can still enhance baseline methods. As shown in Table 4, both clustering difficulty-aware sampling and Fourier perturbation improve SimGCD’s performance even without domain shift.
>
> #### Table 4: performance comparison between SimGCD and our method on CUB (no domain shift)
> |  | CUB All | CUB Old | CUB New |
> |-|-|-|-|
> | SimGCD | 60.3    |  65.6   | 57.7 |
> | FREE | 64.5    | 68.3     | 60.8    |
>
> ### **Q18:computational overhead**
>
> Regarding the computational overhead introduced by FFT, we mitigate this concern by leveraging efficient implementations such as `torch.fft`, which is highly optimized for GPU computation. Additionally, we adopt the FAISS library to significantly accelerate KNN retrieval during training, keeping the extra cost minimal. As for the GMM component, although it relies on effective separation between known and unknown domains, our experiments demonstrate its robustness even in multi-domain scenarios, as shown in Table 6 (Appendix).

---

> > ### Comment · Reviewer_ivLL · 2025-08-04
> >
> > I thank the authors for the rebuttal and for addressing my questions.
> >
> > Regarding my problem with the related work, I believe the coverage is still significantly lacking in both key areas relevant to this paper: domain shift and generalized category discovery (GCD). If the primary focus of the paper is indeed GCD, then it is important to situate the work within that literature. Currently, only five prior works are discussed, and three of those are directly parts of the proposed method, problem setting, and loss functions, and the other two are clear why they have been chosen in particular, while many other more relevant works are missing. For instance, in the context of novel category discovery under domain shift:
> >
> > [1] Boosting Novel Category Discovery Over Domains with Soft Contrastive Learning and All-in-One Classifier (ICCV 2023)
> >
> > It is directly relevant but has not been cited.
> >
> > More critically, foundational GCD methods such as PromptCAL, μGCD, and SPTNet are not mentioned or discussed at all. These are well-known and central to the GCD field, and their absence raises questions about the paper’s background research. Including such references would not only strengthen the paper but also help readers understand the related works and how this work compares or differentiates itself.
> > If these omissions are not addressed, I find it difficult to see how this paper has generalized category discovery as a keyword. I strongly encourage the authors to include a more thorough discussion of relevant prior work to provide the necessary context and clarity for the reader. Here is a non-comprehensive list of recent GCD works, which shows just how many related works are missing in this paper.
> >
> > [2] Learning semi-supervised gaussian mixture models for generalized category discovery (ICCV 2023)
> >
> > [3] Prompt-cal: Contrastive affinity learning via auxiliary prompts for generalized novel category discovery. (CVPR 2023)
> >
> > [4] Dynamic conceptional contrastive learning for generalized category discovery. (CVPR 2023)
> >
> > [5] Discover and align taxonomic context priors for open-world semi-supervised learning (NeurIPS 2023)
> >
> > [6] Learn to categorize or categorize to learn? self-coding for generalized category discovery (NeurIPS 2023)
> >
> > [7] No representation rules them all in category discovery (NeurIPS 2023)
> >
> > [8] SPTNet: An efficient alternative framework for generalized category discovery with spatial prompt tuning (ICLR 2024)
> >
> > [9] Solving the Catastrophic Forgetting Problem in Generalized Category Discovery (CVPR 2024)
> >
> > [10] Contrastive mean-shift learning for generalized category discovery (CVPR 2024)
> >
> > [11] SelEx: Self-expertise in fine-grained generalized category discovery (ECCV 2024)
> >
> > [12] Flipped classroom: Aligning teacher attention with student in generalized category discovery (NeurIPS 2024)
> >
> > [13] Cipr: An efficient framework with cross-instance positive relations for generalized category discovery (TMLR 2024)
> >
> > [14] Debgcd: Debiased learning with distribution guidance for generalized category discovery (ICLR 2025)
> >
> >
> >
> > This is not also only limited to GCD, also for domain shift with Fourier many relevant works are missing:
> >
> > [15] A Fourier-based Framework for Domain Generalization (CVPR 2021)
> >
> > [16] Domain Generalization via Frequency-domain-based Feature Disentanglement and Interaction (ACM Multimedia 2022)
> >
> > [17] Domain Generalization with Fourier Transform and Soft Thresholding (ICASSP 2024)
> >
> > Overall, given the non-comprehensive related work, the complex setup, and several issues with the clarity and consistency of the text (which seems to be a not carefully reviewed output of AI tools), I believe it is essential that the authors address these major concerns in the final version. Additionally, I was unable to find the actual code in the Anonymous GitHub link provided in the submission; only the paper’s abstract is available in that link. Combined with unresolved formula inconsistencies, limited related work coverage, and the inclusion of a non-functional code link, this raises some concerns about the reliability and reproducibility of the reported experimental results. In my opinion, a public code release for this paper is critical to ensure transparency and reproducibility.
> >
> > For now, I will maintain my initial score, but I strongly encourage the authors to take extra care in the reporting and writing of the paper, and to thoroughly double-check all aspects of the submission to ensure clarity and accuracy. Especially if possible, they should update the anonymous GitHub link with the actual code.

---

> > > ### Author Response · Authors · 2025-08-05
> > >
> > > We sincerely thank the reviewer for the thoughtful and detailed feedback. We address the raised concerns below:
> > >
> > > 1. On Related Work Coverage
> > > We appreciate the reviewer’s suggestions and acknowledge that the related work section in the current submission could benefit from a more comprehensive discussion of recent advances in Generalized Category Discovery (GCD) and domain shift. In the final version, we will incorporate the suggested references into the related work section to better situate our contributions within the broader GCD literature. Our primary baseline is built upon SimGCD, which we view as a strong and representative method for the Generalized Category Discovery (GCD) setting. Based on this, our main comparative analyses have focused on SimGCD and its close variants.
> > >
> > > Moreover, we clarify that we have already compared our method against several of the most recent and competitive GCD methods in our experimental section, including Contrastive Mean-Shift (CMS), SPTNet, and ProtoGCD, as also noted in our response to Reviewer Jovx. Our results consistently demonstrate that these state-of-the-art GCD approaches fail to generalize effectively in our more challenging setting, while our method exhibits superior performance.
> > >
> > > Due to space constraints, we could not elaborate on all GCD methods in the main paper. However, we will include detailed discussions of additional GCD works such as μGCD, PromptCAL, SelEx, DEBGCD, and Flipped-Classroom, etc., in the supplementary material.
> > >
> > > 2. Regarding the Paper “Boosting Novel Category Discovery Over Domains with Soft Contrastive Learning and All-in-One Classifier”
> > > Thank you for pointing out this paper. We have carefully reviewed it and would like to clarify our position:
> > >
> > > Although the title includes “category discovery,” the paper does not belong to the Generalized Category Discovery (GCD) setting. It focuses on open-set domain adaptation with a strong emphasis on improving closed-set classification and separating known vs. unknown data, but it does not tackle the discovery of novel categories in unlabeled data nor does it evaluate on standard GCD benchmarks.
> > >
> > > By contrast, our work specifically addresses the discovery of new categories from mixed-domain unlabeled data, consistent with the GCD definition in prior works. For this reason, we chose to compare against methods like HiLo and CDAD-Net, which, despite some differences in assumptions (e.g., unlabeled data from unknown domains only for CDAD-Net), are much more aligned with our problem setup. In both cases, our method demonstrates stronger performance.
> > >
> > >
> > > 3. Regarding the Anonymous GitHub Link
> > > We apologize for the earlier incomplete GitHub repository. Thank you for pointing this out. We have now uploaded the full codebase to the anonymous GitHub repository as required. The repository includes all necessary scripts, configurations, and instructions to reproduce the main results reported in the paper.
> > >
> > > 4. Clarity and Consistency in the Writing
> > > We appreciate the reviewer’s note on textual clarity. We have conducted an extensive review of the manuscript to eliminate any inconsistencies or unclear sections. The final version will incorporate a polished presentation of the technical content and clear mathematical exposition.
> > >
> > > We thank the reviewer again for their valuable feedback. We believe the above clarifications and planned updates will address the concerns raised and improve the overall quality, transparency, and completeness of our submission.

---

> > > > ### Comment · Reviewer_ivLL · 2025-08-05
> > > >
> > > > I thank the authors for providing a detailed response and for addressing all my remaining concerns, including the improvements to text clarity and consistency, a better positioning of the paper regarding related works, and additional experiments suggested by other reviewers. I especially appreciate the release of the codebase, even though I have not carefully analyzed it yet; making the source code available increases confidence in the work's reproducibility.
> > > >
> > > > With all my concerns addressed, I am in favor of accepting this paper.

---

### Official Review · Reviewer_rRLy · 2025-07-01

**Clarity:** 3
**Significance:** 3
**Originality:** 3
**Rating:** 4
**Confidence:** 4

**Summary:**

This paper introduces *Frequency-guided Generalized Category Discovery (Free)*, a novel framework for Domain-Shifted Generalized Category Discovery (DS\_GCD), which aims to categorize both known and unknown classes from unlabeled data spanning both seen and novel domains. In contrast to existing approaches, *Free* takes a frequency-domain perspective by leveraging the amplitude and phase of the Fourier transform to model style and semantic content, respectively. Experiments on DomainNet and the newly proposed SSB-C benchmark demonstrate that *Free* achieves superior results compared to prior GCD and NCD methods.

**Questions:**

Please refer to the weakness part.

**Ethical Concerns:**

["NO or VERY MINOR ethics concerns only"]

**Final Justification:**

The authors have addressed most of my concerns.

**Limitations:**

Please refer to the weakness part.

**Paper Formatting Concerns:**

No paper formatting concerns

**Quality:**

3

**Strengths And Weaknesses:**

**Strengths**

1. The proposed *Free* framework consistently outperforms a wide range of state-of-the-art methods on challenging domain-shifted benchmarks.
2. The paper is clearly written and well-structured, making it accessible and easy to follow.

**Weaknesses**

1. While the use of Fourier-based techniques is innovative in the GCD context, such methods are well-studied in Domain Adaptation (DA) and Domain Generalization (DG). Given that the datasets are well-examined in DA/DG literature, it is not entirely surprising that *Free* achieves improved results.
2. Although the paper explains how frequency manipulation may help mitigate domain shifts, it lacks a deeper analysis of how such manipulation specifically aids the discovery of unknown classes. For example, does the Intra-Domain Frequency Perturbation (IDFP) affect intra-class compactness or inter-class separability?
3. The *Free* framework comprises four main components, resulting in a relatively complex pipeline. The reliance on Fourier transforms and extensive data augmentations may introduce considerable computational overhead. The paper does not provide an analysis of this computational cost (e.g., training time, FLOPs) relative to the baseline methods.
4. Minor issue: Lines 458, 460, 463, and 465 contain Markdown syntax.

---

> ### Author Rebuttal · Authors · 2025-07-30
>
> ### **Q1:contribution of each component:**
> We appreciate the reviewers’ recognition of the practical value and challenges associated with our proposed problem setting. Indeed, our formulation—which combines labeled and unlabeled data from distinct domains and categories for category discovery—is a natural yet underexplored generalization of UDA and GCD. While prior work has tackled GCD under domain homogeneity or UDA under closed-label space, few have addressed both semantic and domain shift simultaneously. Our work is to bridge this gap with a unified and principled solution. While we acknowledge that some components draw inspiration from previous work, we would like to emphasize that our method is not a trivial composition of existing modules, but rather introduces key innovations specifically tailored to the dual-challenge of domain and category shifts. First, we observed that directly applying previous frequency-based adaptation methods (e.g., FDA) to the GCD setting is problematic, as randomly mixing frequency components from different domains may introduce spurious correlations and semantic noise. This is especially detrimental for novel categories, which rely solely on pseudo-label supervision. To address this, we propose an FDS mechanism that explicitly separates samples by domain before applying perturbation, preventing harmful signal contamination and ensuring more stable representation learning and better discrimination between known and novel categories. Building on this, our class-aware cross-domain frequency perturbation uses the more familiar source domain data to guide feature learning and mitigate domain gaps without disrupting category structure. To further improve the model’s robustness within the unknown domain, we introduce an intra-domain frequency perturbation module. This enhances the model’s invariance to intra-domain variations, addressing the lack of supervision and diversity in unlabeled novel categories. Existing GCD or UDA approaches typically assume uniform difficulty across categories during training. However, in practice, some categories (especially in the unknown domain) are harder to cluster due to greater intra-class variance or domain shifts. Our proposed CDAS module adaptively reweights the sampling process based on per-category clustering difficulty, allowing the model to focus its capacity on harder categories, improving both representation learning and final clustering quality. To our knowledge, this is the first work to incorporate such difficulty-awareness into domain-adaptive category discovery.
> As shown in Table 1, directly applying frequency transformations without domain separation often leads to negative transfer, especially in the unseen domain where pseudo-labels are noisy. This is because the model relies entirely on pseudo-labels to supervise clustering in these unfamiliar domains, which may result in noisy guidance and degraded performance.
> This motivated us to first perform domain separation and leverage the more familiar source domain to learn domain-invariant feature representations. Then, we apply In-Domain Frequency Perturbation (IDFP) within the target domain to avoid collapse of semantic structures.
> As shown in Table 2, IDFP consistently improves performance across all/known/unknown classes in the unseen domain, indicating that it helps enhance intra-class compactness and inter-class separability. Moreover, the Cross-Domain Frequency Perturbation (CDFP) significantly improves generalization, suggesting that it promotes learning of domain-invariant representations.
>
> #### **Table 1: Ablation study on different transformation strategies**
>
> | Method     | Real All | Real Old | Real New | Painting All | Painting Old | Painting New |
> |------------|----------|----------|----------|---------------|---------------|----------------|
> | SimGCD     | 61.3       | 77.8       | 52.9       | 34.5      | 35.6      | 33.5   |
> | Random     | 62.7     | 76.5     | 54.3     | 33.2          | 34.5          | 32.5           |
> | FREE  | 67.7     | 78.1     | 61.2     | 45.6          | 46.1          | 44.8           |
>
> #### **Table 2: Ablation study of different components.**
>
> | FDS | IDFP | CDFP | CDAS | Real All | Real Old | Real New | Painting All | Painting Old | Painting New |
> |:---:|:----:|:----:|:----:|:--------:|:--------:|:--------:|:-------------:|:------------:|:-------------:|
> | ✗   | ✗    | ✗    | ✗    | 61.3     | 77.8     | 52.9     | 34.5          | 35.6         | 33.5          |
> | ✓   | ✓    | ✗    | ✗    | 62.0     | 77.8     | 53.3     | 36.6          | 37.8         | 35.5          |
> | ✓   | ✗    | ✓    | ✗    | 65.6     | 77.9     | 58.2     | 41.6          | 41.9         | 40.7          |
> | ✓   | ✗    | ✗    | ✓    | 63.1     | 77.9     | 54.8     | 37.6          | 38.0         | 36.9          |
> | ✓   | ✓    | ✓    | ✗    | 66.1     | 78.0     | 59.5     | 42.8          | 43.1         | 42.0          |
> | ✓   | ✓    | ✓    | ✓    | 67.7     | 78.1     | 61.2     | 45.6          | 46.1         | 44.8          |
>
> ### **Q2:Computational Complexity**
> We evaluate the computational efficiency of our method relative to baseline approaches. Although our framework introduces additional operations—namely spectral decomposition via Fast Fourier Transform (FFT), K-nearest neighbor similarity estimation, and Gaussian Mixture Model (GMM)–based domain modeling—these are implemented using highly optimized tools. Specifically, we utilize torch.fft for frequency transformations and FAISS for efficient KNN retrieval on GPU.
> As shown in the table below, we report the FLOPs of a single forward step and the elapsed time per iteration on the CUB-C dataset with a batch size of 128 on an NVIDIA A100 GPU. Our method maintains a comparable computational profile in terms of both FLOPs and per-iteration training time:
>
> #### **Table 3: FLOPs and training time cost**
>
> | Method  | FLOPs (G) | Time Cost per Iteration (s) |
> |---------|-----------|------------------------------|
> | SimGCD  | 2159.3    | 1.079    |
> | FREE | 2160.8    | 1.132 |
>
> These results demonstrate that the proposed enhancements introduce minimal overhead, while enabling more effective domain-aware learning.
>
> ### **Q3:Markdown syntax**
> Thanks for the suggestion. We will properly fix it in our final version.

---

> > ### Comment · Reviewer_rRLy · 2025-08-06
> >
> > I appreciate the authors for their detailed responses. Most of my concerns have been addressed. After carefully reviewing the responses and their rebuttal to other reviewers, I will keep my positive assessment. I suggest the authors add these additional experiments and explanations to the final version of the paper.

---

> > > ### Author Response · Authors · 2025-08-09
> > >
> > > We sincerely thank you for your thoughtful, constructive, and encouraging feedback throughout the review process, which has greatly improved the clarity and depth of our work. It is truly a privilege to have our work evaluated by someone with such discernment and scholarly acumen. We are delighted that your remaining concerns have been resolved, and if you feel confident to update your score or confidence rating, we would be deeply grateful.
> > >
> > > As a clarification, the ablation studies in our rebuttal Table 1 and Table 2 correspond to Table 3 and Table 4 in the main paper. In the final version, we will also include additional complexity analysis, and further refine the introduction and method sections to better emphasize our motivation and design rationale. Thank you again for your valuable time and insight.

---

> ### Comment · Area_Chair_you4 · 2025-08-05
>
> Dear Reviewer rRLy,
>
> The author-rebuttal phase is now underway, and the authors have provided additional clarifications and performance results in their rebuttal. Could you please take a moment to review their response and engage in the discussion? In particular, we’d appreciate your thoughts on whether their revisions adequately address your initial concerns. Thank you for your time and valuable contributions.
>
> Best, Your AC

---

### Official Review · Reviewer_Jovx · 2025-07-03

**Clarity:** 3
**Significance:** 2
**Originality:** 3
**Rating:** 4
**Confidence:** 5

**Summary:**

This paper introduces a novel framework, "Free," to tackle Domain-Shifted Generalized Category Discovery (DS-GCD), a challenging setting where unlabeled data contains both new categories and samples from unseen domains. The method is built on the core insight that an image's Fourier amplitude spectrum captures domain-specific style, while the phase spectrum retains semantic content. "Free" leverages this by first probabilistically separating samples into known and unknown domains based on amplitude similarity. It then employs class-aware frequency perturbation strategies to align distributions and enhance robustness, further guided by a difficulty-aware sampling mechanism. Extensive experiments on DomainNet and SSB-C benchmarks show that the proposed method significantly outperforms prior state-of-the-art approaches.

**Questions:**

1. Regarding Baselines: The paper's motivation hinges on the premise that standard GCD methods struggle under domain shift. To make this point more compelling, could you add more recent and powerful GCD methods like ProtoGCD [1], CMS [2], and SPTNet [3] into your main table? Demonstrating that these stronger methods also fail in the DS-GCD setting would significantly strengthen the paper's central motivation.
2. Regarding Hyperparameter Sensitivity: The proposed 'Free' framework is methodologically complex. While the sensitivity analysis in Section B.7 is helpful, it appears to omit a key hyperparameter. Could you please elaborate on the model's sensitivity to the confidence threshold $\mu$ used in CDFP? A more complete analysis would increase confidence in the method's robustness.
3. Regarding Statistical Significance: The main results in Tables 1 and 2 report the mean accuracy over three runs but omit standard deviations or other measures of variance. Could you provide these values? This information is crucial for assessing the statistical significance and stability of the reported performance gains, especially for a complex, multi-component method.

[1] Protogcd: Unified and unbiased prototype learning for generalized category discovery

[2] Contrastive mean-shift learning for generalized category discovery.

[3] Sptnet: An efficient alternative framework for generalized category discovery with spatial prompt tuning.

**Ethical Concerns:**

["NO or VERY MINOR ethics concerns only"]

**Final Justification:**

The authors’ detailed rebuttal has addressed my remaining questions. This confirms my initial positive assessment, and therefore, my score remains unchanged.

**Limitations:**

yes

**Paper Formatting Concerns:**

The paper is well-written.

**Quality:**

3

**Strengths And Weaknesses:**

Strengths:
* S1: Novel and Effective Approach for an Important Problem. The paper tackles the highly relevant and challenging problem of Generalized Category Discovery under a domain shift (DS-GCD). The core technical contribution—using the Fourier amplitude and phase spectra to disentangle domain-specific style from semantic content—is an elegant, well-motivated, and novel approach.
* S2: State-of-the-Art Empirical Performance. The empirical evaluation is extensive and demonstrates state-of-the-art performance. The proposed 'Free' method consistently and substantially outperforms all baselines, including the specialized DS-GCD method HiLo, across the challenging DomainNet and SSB-C benchmarks (Tables 1 & 2). The magnitude of the gains convincingly validates the method's effectiveness.
* S3: High-Quality and Rigorous Presentation. The paper is well-written and easy to follow. The authors' claims are supported by detailed ablation studies (Table 3) that clearly justify the contribution of each component. The inclusion of a comprehensive appendix with additional analyses further strengthens the quality and completeness of the work.

Weaknesses:
* W1: Limited Comparison to Recent GCD Methods. A key premise of the paper is that standard GCD methods are not suitable for the DS-GCD setting. However, the baselines used to demonstrate this (e.g., GCD, SimGCD) are not the most recent. Including stronger, contemporary GCD methods (such as ProtoGCD [1], CMS [2], or SPTNet [3]) and showing that they also struggle under domain shift would provide a more powerful motivation for the necessity of a specialized approach like Free.
* W2: Incremental Novelty of Some Components. While the overall framework is novel, the novelty of some individual components is limited. For instance, the frequency-based perturbation is a well-designed, class-aware extension of the technique from FDA[4], and the clustering-difficulty-aware sampling is a relatively common strategy for handling hard examples. The paper would be strengthened by a clearer discussion of these connections.
* W3: High Methodological Complexity and Incomplete Analysis. The proposed framework is composed of four main modules, resulting in a system with high methodological complexity and lots of hyperparameters that could be a practical drawback for adoption. The hyperparameter sensitivity analysis in the appendix is appreciated but incomplete; it omits several key parameters introduced by the method, such as the confidence threshold $\mu$. A more complete analysis would increase confidence in the method's robustness.

[1] Protogcd: Unified and unbiased prototype learning for generalized category discovery

[2] Contrastive mean-shift learning for generalized category discovery.

[3] Sptnet: An efficient alternative framework for generalized category discovery with spatial prompt tuning.

[4] Fda: Fourier domain adaptation for semantic segmentation.

---

> ### Author Rebuttal · Authors · 2025-07-30
>
> ### **Q1:Regarding Baselines:**
> We have included comparisons with recent GCD methods, including ProtoGCD, CMS, and SPTNet, as shown in Table 1 and Table 2. While these methods bring some improvements, they still struggle under domain shift. In contrast, our method consistently outperforms them, further validating the need for a specialized solution like Free.
>
> ### Table 1: Performance on the SSB-C Dataset
> | Method     | CUB-O All | CUB-O Old | CUB-O New | CUB-C All | CUB-C Old | CUB-C New | Scars-O All | Scars-O Old | Scars-O New | Scars-C All | Scars-C Old | Scars-C New | FGVC-O All | FGVC-O Old | FGVC-O New | FGVC-C All | FGVC-C Old | FGVC-C New |
> |-|-|-|-|-|-|-|-|-|-|-|-|-|-|-|-|-|-|-|
> | ProtoGCD   | 33.2±1.8  | 35.9±1.3  | 32.3±2.2  | 31.7±1.9  | 33.8±1.5  | 27.2±2.3  | 29.1±1.4     | 41.9±2.0     | 28.2±1.6     | 24.8±2.1     | 33.6±1.7     | 16.4±1.9     | 29.8±2.0    | 31.5±1.1    | 28.2±2.2    | 25.7±1.6    | 26.6±2.4    | 24.6±1.5    |
> | CMS        | 33.0±1.7  | 34.5±1.1  | 31.2±1.9  | 30.1±2.0  | 33.1±1.4  | 26.1±2.2  | 28.0±1.5     | 40.2±2.0     | 27.9±1.6     | 24.2±2.3     | 32.1±1.8     | 16.3±1.3     | 28.7±2.0    | 30.2±1.7    | 27.9±2.2    | 24.8±1.5    | 25.7±1.1    | 23.9±2.4    |
> | SPTNet     | 35.9±1.9  | 35.1±1.2  | 33.2±2.1  | 32.3±1.4  | 34.8±2.0  | 28.5±1.1  | 29.8±1.8     | 41.2±2.2     | 30.4±1.5     | 25.3±1.0     | 33.4±2.1     | 18.1±1.6     | 27.9±1.3    | 30.1±2.3    | 26.8±1.4    | 24.4±2.2    | 26.8±1.2    | 22.7±2.0    |
> | FREE | 60.4±1.1  | 58.5±1.4  | 63.2±1.3  | 55.7±1.6  | 57.1±1.7  | 53.7±1.5  | 43.6±1.1     | 48.1±1.5     | 40.8±1.2     | 38.9±1.7     | 46.1±1.6     | 32.6±1.5     | 48.5±1.3    | 54.9±1.4    | 51.2±1.1    | 35.0±1.5    | 32.4±1.3    | 38.9±1.2    |
> ### Table 2a: Real+Painting Performance on DomainNet
> | Method     | Real All       | Real Old       | Real New       | Painting All   | Painting Old   | Painting New   |
> |-|-|-|-|-|-|-|
> | ProtoGCD   | 61.8±1.5       | 77.3±1.4       | 54.9±2.2       | 36.8±1.5       | 36.5±1.4       | 35.6±1.8       |
> | CMS        | 61.6±1.5       | 76.9±2.3       | 54.7±2.0       | 35.2±1.8       | 35.9±1.2       | 35.1±1.2       |
> | SPTNet     | 62.1±1.9       | 78.3±1.2       | 53.8±1.1       | 36.9±2.3       | 35.7±2.4       | 36.2±2.1       |
> | FREE | 67.7±1.4       | 78.1±1.7       | 61.2±1.2       | 45.6±1.1       | 46.1±1.3       | 44.8±1.6       |
> ### Table 2b: Real+Sketch Performance on DomainNet
> | Method     | Real All       | Real Old       | Real New       | Sketch All   | Sketch Old   | Sketch New   |
> |-|-|-|-|-|-|-|
> |ProtoGCD   | 63.1±1.5       | 77.9±1.6       | 55.2±2.1      | 17.8±1.2       | 21.5±1.3       | 14.5±1.3    |
> | CMS      | 63.3±1.1        | 77.8±2.2        | 55.3±1.8        | 16.7±1.0        | 26.0±1.0        | 11.3±1.4        |
> | SPTNet   | 62.8±1.4        | 77.4±1.1        | 55.7±2.0        | 17.0±1.4       | 20.4±1.2        | 15.2±1.1        |
> | FREE | 67.8±1.5        | 78.2±1.4        | 61.6±1.3        | 22.5±1.4        | 25.8±1.3        | 20.9±1.1        |
> ### Table 2c: Real+Quickdraw Performance on DomainNet
> | Method     | Real All       | Real Old       | Real New       |Quickdraw All   | Quickdraw Old   | Quickdraw New   |
> |-|-|-|-|-|-|-|
> | ProtoGCD   | 51.3±1.6       | 69.2±1.7       | 40.2±2.5       | 7.2±0.9       | 6.1±0.8       | 8.1±0.8       |
> | CMS      | 47.1±2.2        | 65.6±1.3        | 35.4±1.3        | 6.9±1.0         | 5.7±1.1         | 7.7±0.7         |
> | SPTNet   | 49.1±1.0        | 67.8±2.3        | 38.0±1.4        | 7.0±1.1         | 5.8±0.6         | 7.8±0.8         |
> | FREE | 61.4±0.9        | 78.1±1.6        | 55.1±1.8        | 8.9±0.7         | 7.8±0.5        | 9.0±0.9         |
> ### Table 2d: Real+Clipart Performance on DomainNet
> | Method     | Real All       | Real Old       | Real New       | Clipart All   | Clipart Old   | Clipart New   |
> |-|-|-|-|-|-|-|
> | ProtoGCD   | 63.1±1.3       | 77.3±1.7       | 55.8±2.4       | 25.8±1.6       | 38.7±1.4       | 14.8±1.5       |
> | CMS    | 62.5±1.6        | 76.5±1.4        | 55.4±1.9        | 24.7±1.2        | 30.9±1.4        | 18.8±1.5        |
> | SPTNet   | 62.3±1.8        | 77.1±1.3        | 54.7±2.4        | 24.5±2.1        | 38.0±2.3        | 13.9±2.3        |
> | FREE | 66.4±1.0        | 78.1±0.9        | 60.1±0.9        | 29.3±1.5        | 37.2±1.6        | 26.3±1.6        |
> ### Table 2e: Real+Infograph Performance on DomainNet
> | Method     | Real All       | Real Old       | Real New       | Infograph All   | Infograph Old   |Infograph New   |
> |-|-|-|-|-|-|-|
> | ProtoGCD   | 55.6±2.4       | 68.8±1.7       | 46.8±2.6       | 12.8±1.7       | 16.6±1.5       | 9.9±1.2       |
> | CMS      | 54.5±1.6        | 67.9±2.1        | 46.2±1.3        | 11.9±1.7        | 19.4±1.8        | 7.9±1.1         |
> | SPTNet   | 57.2±1.8        | 68.3±2.3        | 48.1±1.1        | 12.0±1.3        | 15.9±1.1        | 9.8±1.5         |
> | FREE| 68.1±1.4        | 78.9±1.3        | 60.2±1.6        | 16.1±1.4        | 18.6±1.2        | 13.4±0.9        |
>
> ### **Q2:Regarding contribution of each component:**
> We appreciate the reviewers’ recognition of the practical value and challenges associated with our proposed problem setting. Indeed, our formulation—which combines labeled and unlabeled data from distinct domains and categories for category discovery—is a natural yet underexplored generalization of UDA and GCD. While prior work has tackled GCD under domain homogeneity or UDA under closed-label space, few have addressed both semantic and domain shift simultaneously. Our work is to bridge this gap with a unified and principled solution. While we acknowledge that some components draw inspiration from previous work, we would like to emphasize that our method is not a trivial composition of existing modules, but rather introduces key innovations specifically tailored to the dual-challenge of domain and category shifts.
>
> First, we observed that directly applying previous frequency-based adaptation methods (e.g., FDA) to the GCD setting is problematic, as randomly mixing frequency components from different domains may introduce spurious correlations and semantic noise. This is especially detrimental for novel categories, which rely solely on pseudo-label supervision. To address this, we propose an FDS mechanism that explicitly separates samples by domain before applying perturbation, preventing harmful signal contamination and ensuring more stable representation learning and better discrimination between known and novel categories. Building on this, our class-aware cross-domain frequency perturbation uses the more familiar known-domain data to guide feature learning and mitigate domain gaps without disrupting category structure. To further improve the model’s robustness within the unknown domain, we introduce an intra-domain frequency perturbation module. This enhances the model’s invariance to intra-domain variations, addressing the lack of supervision and diversity in unlabeled novel categories. Existing GCD or UDA approaches typically assume uniform difficulty across categories during training. However, in practice, some categories (especially in the unknown domain) are harder to cluster due to greater intra-class variance or domain shifts. Our proposed CDAS module adaptively reweights the sampling process based on per-category clustering difficulty, allowing the model to focus its capacity on harder categories, improving both representation learning and final clustering quality. To our knowledge, this is the first work to incorporate such difficulty-awareness into domain-adaptive category discovery.
>
> ### **Q3:Regarding Hyperparameter Sensitivity:**
> We have further conducted a parameter analysis on \\(\eta\\), as shown in Table 3. We observe that smaller \\(\eta\\) values tend to introduce noisy pseudo-labels and degrade performance, while larger \\(\eta\\) can effectively filter out noisy samples, leading to more accurate category-aware perturbation.
> ### Table 3: Parameter Analysis of \\(\eta\\)
>
> | \\(\eta\\)    | Real All | Real Old | Real New | Painting All | Painting Old | Painting New |
> |-|-|-|-|-|-|-|
> | 0. 1  | 65.6     | 77.6     | 58.3    | 43.5         | 44.6          | 42.1         |
> | 0.5 | 66.2     | 77.8     | 59.1    | 44.1          | 45.2           | 42.8           |
> | 0.9  | 67.5    | 78.0     | 60.9    | 45.4          | 45.9          | 44.6           |
> | 0.95  | 67.7     | 78.1     | 61.2     | 45.6          | 46.1           | 44.8           |
> ### **Q4:Regarding Statistical Significance:**
> We thank the reviewer for pointing this out. We have added standard deviation results in Table 1 and Table 2, which show the performance is stable across runs. The error bars for other methods, such as GCD, SimGCD, and ORCA, can be referenced from Appendix Figure 8 in the HiLo paper. Due to space limitations, we will include the full error statistics in the final version.

---

> > ### Comment · Reviewer_Jovx · 2025-08-05
> >
> > I have read the authors' rebuttal. I thank them for adding new experiments to address my concerns.
> >
> > Resolved Issues:
> >  * The authors added stronger baselines (ProtoGCD, CMS, SPTNet), which strengthens the paper.
> >  * Standard deviations were added to the results, improving rigor.
> >
> > Remaining Weakness:
> >  * The key issue is the incomplete hyperparameter analysis for $\eta$. The new analysis shows performance improves up to the chosen value (0.95) and stops. This makes the parameter choice seem arbitrary and does not sufficiently prove robustness.
> >
> > Due to the remaining weakness, my score remains “Borderline Accept”.

---

> ### Author Response · Authors · 2025-08-05
>
> We thank the reviewer for their constructive feedback and for acknowledging the improvements made to the manuscript.
>
> Regarding the concern about the hyperparameter $\eta$, we would like to clarify that our choice of a relatively high threshold value (e.g., $\eta = 0.95$) is not arbitrary, but rather guided by empirical intuition. Specifically, $\eta$ controls the confidence level for selecting unlabeled samples used in frequency-domain amplitude exchange. A lower value of $\eta$ may introduce samples with incorrect pseudo-labels, resulting in semantic misalignment during the style exchange process and deteriorated clustering performance.
>
> This phenomenon is particularly critical in fine-grained datasets. For instance, bird images are often associated with sky-like backgrounds, while fish typically appear in underwater scenes. Mixing frequency characteristics across such semantically distinct categories can introduce domain-specific noise that impairs the model's ability to discover meaningful clusters.
>
> Therefore, using a higher $\eta$ (e.g., 0.95) helps us filter out low-confidence samples and preserve the semantic integrity of style representations. As shown in our hyperparameter analysis, performance tends to improve as $\eta$ increases, and stabilizes around the chosen value, indicating that our framework is not overly sensitive to this parameter while still benefiting from confident sample selection.
>
> To further address your concern, we extended the range of the hyperparameter $\eta$ and observed that model performance tends to stabilize as $\eta$ increases. This supports our choice of setting $\eta$ to a relatively high value, as it helps filter out low-confidence samples and suppresses semantically noisy style transfer (e.g., background mismatches such as sky in bird images versus water in fish images), which may hurt clustering quality.
>
> ### Table 3: Parameter Analysis of $\eta$
>
> | $\eta$| Real All | Real Old | Real New | Painting All | Painting Old | Painting New |
> |--------:|----------:|----------:|----------:|---------------:|---------------:|---------------:|
> | 0.96    | 67.6     | 78.0     |  61.1  | 45.5          | 46.1          | 44.6          |
> | 0.97   | 67.5     | 78.1     | 60.8    | 45.4          | 46.0         | 44.5          |
>
> We will incorporate this explanation into the final version for better clarity.

---

> > ### Comment · Reviewer_Jovx · 2025-08-06
> >
> > I thank the authors for their detailed response, which has addressed my concerns. I am happy to recommend acceptance of this manuscript.

---

### Note · Authors · 2025-08-14

Dear Area Chair,

We would like to briefly summarize the reviewer feedback for our submission #2023 “Generalized Category Discovery under Domain Shift: A Frequency Domain Perspective.” We greatly appreciate the constructive comments from all reviewers and believe all concerns have been fully addressed.

Jovx (Rating: 4 → Accept)
Recognized our novel and effective DS-GCD approach and strong results on SSB-C and DomainNet. After we added stronger baselines (ProtoGCD, CMS, SPTNet), statistical significance analysis, and hyperparameter sensitivity clarification, Jovx confirmed all concerns were resolved and recommended acceptance.

rRLy (Rating: 4 → Accept)
Praised the strength and clarity of our method. Following our additional ablations, computational complexity analysis, and detailed component contribution explanations, rRLy maintained a positive view and encouraged including these updates in the final version.

ivLL (Rating: 4 → Accept)
Initially raised concerns about related work and code availability. We expanded the related work, released the complete anonymous code, and clarified our positioning in the GCD literature. ivLL expressed satisfaction and confirmed support for acceptance.

xWh6 (Rating: 3 → Borderline Accept)
Raised questions on motivation clarity and missing CDAD-Net comparison. We added CDAD-Net results (~20% improvement), strengthened the motivation with earlier placement of key analyses, and clarified the rationale for each module. The reviewer acknowledged these improvements, agreed they enhanced the paper, and suggested highlighting them earlier in the final version.

Summary
Three reviewers updated to Accept, and the fourth to Borderline Accept with positive remarks. Our key contributions include:

The first frequency-domain framework for DS-GCD

Novel modules — Frequency-based Domain Separation, Class-aware Cross-/Intra-Domain Frequency Perturbations, and Clustering Difficulty-Aware Sampling — addressing semantic noise, domain gaps, and uneven cluster separability

Consistent, significant gains over state-of-the-art GCD and DS-GCD methods across multiple challenging benchmarks

We hope this strong positive consensus will be reflected in the final decision. We will ensure all additional experiments, clarifications, and analyses appear in the camera-ready version.

Best regards,
The Authors

---

### Decision · Program_Chairs · 2025-09-17

**Decision:**

Accept (poster)

**Comment:**

This paper introduces a novel frequency-based framework for Domain-Shifted Generalized Category Discovery (DS-GCD) that leverages the Fourier spectrum to separate domain-specific style (amplitude) from semantic content (phase). The method employs class-aware frequency perturbations and a difficulty-aware sampling mechanism, achieving state-of-the-art performance on the DomainNet and SSB-C benchmarks.

The reviewers acknowledged its strengths including its novelty in the GCD problem setting, strong empirical performance across challenging benchmarks, and well-structured presentation.  However, several major concerns were raised: 1) limited novelty in some components, which adapt existing techniques from domain adaptation literature without significant innovation; 2) high complexity with many hyperparameters, lacking thorough parameter sensitivity analysis; 3) missing comparisons with important recent work like CDAD-Net and lacking proper citations for relevant prior art; and 4) insufficient analysis of mechanisms: The paper doesn't adequately explain how frequency manipulation specifically aids unknown class discovery or justify design choices like using GMM for domain separation.

In their rebuttal, the authors provided a detailed response, including clarification on differentiation from domain adaptation techniques, computational complexity results, additional ablation studies, comparisons with recent methods and hyperparameter analysis. All four reviewers positively responded to the rebuttal, with Reviewer xWh6 raising their ratings to borderline accept and the other reviewers maintaining their borderline accept ratings. The AC concurs with the reviewers' assessment, noting that while the added experiments and clarifications from the rebuttal stage are substantial, they have addressed most concerns and can be incorporated in revision. Overall, the main merits of the work outweigh its weaknesses and therefore the AC recommends acceptance. The authors should revise the manuscript to incorporate the reviewers' feedback and address the points discussed in the rebuttal.